# NRF2 regulates core and stabilizing circadian clock loops, coupling redox and timekeeping in *Mus musculus*

Ryan S Wible[1,2†], Chidambaram Ramanathan[3†], Carrie Hayes Sutter[2,3], Kristin M Olesen[3], Thomas W Kensler[4], Andrew C Liu[2,3†], Thomas R Sutter[1,2,3†]*

[1]Department of Chemistry, University of Memphis, Memphis, United States; [2]W Harry Feinstone Center for Genomic Research, University of Memphis, Memphis, United States; [3]Department of Biological Sciences, University of Memphis, Memphis, United States; [4]Department of Pharmacology and Chemical Biology, University of Pittsburgh, Pittsburgh, United States

*For correspondence:
tsutter@memphis.edu

[†]These authors contributed equally to this work

**Competing interests:** The authors declare that no competing interests exist.

**Abstract** Diurnal oscillation of intracellular redox potential is known to couple metabolism with the circadian clock, yet the responsible mechanisms are not well understood. We show here that chemical activation of NRF2 modifies circadian gene expression and rhythmicity, with phenotypes similar to genetic NRF2 activation. Loss of *Nrf2* function in mouse fibroblasts, hepatocytes and liver also altered circadian rhythms, suggesting that NRF2 stoichiometry and/or timing of expression are important to timekeeping in some cells. Consistent with this concept, activation of NRF2 at a circadian time corresponding to the peak generation of endogenous oxidative signals resulted in NRF2-dependent reinforcement of circadian amplitude. In hepatocytes, activated NRF2 bound specific enhancer regions of the core clock repressor gene *Cry2*, increased *Cry2* expression and repressed CLOCK/BMAL1-regulated E-box transcription. Together these data indicate that NRF2 and clock comprise an interlocking loop that integrates cellular redox signals into tissue-specific circadian timekeeping.
DOI: https://doi.org/10.7554/eLife.31656.001

## Introduction

Circadian clocks are an evolutionarily conserved timekeeping mechanism that allows organisms to anticipate and adapt their behavior, physiology, and biochemistry to predictable changes in their environment. Organisms with a circadian period length matched to that of their environment grow faster and have greater reproductive fitness and lifespan than their unsynchronized competitors (*Dodd et al., 2005*; *Woelfle et al., 2004*; *Ouyang et al., 1998*; *Beaver et al., 2002*)

Circadian timekeeping is a product of a hierarchical system composed of multiple oscillators (*Reppert and Weaver, 2002*). The suprachiasmatic nuclei (SCN) of the anterior hypothalamus represent the top of the hierarchy, responsible for integrating environmental light-dark input and synchronizing timekeeping in peripheral tissues (*Liu et al., 2007a*). Most tissues possess autonomous cellular oscillators that control rhythmic tissue-specific physiology. At the molecular level, the circadian clock is composed of a core transcription-translation feedback loop, in which transcriptional activators aryl hydrocarbon receptor nuclear translocator-like (ARNTL/BMAL1) and circadian locomotor output cycles kaput (CLOCK) regulate the transcription of their own repressors, period 1/2/3 (*Per1/2/3*) and cryptochrome 1/2 (*Cry1/2*) (*Ko and Takahashi, 2006*; *Reppert and Weaver, 2002*). PER and CRY repress their own transcription through the inhibition of the CLOCK/BMAL1 transcriptional complex forming the core feedback loop (*Zhang and Kay, 2010*). The fidelity of the core clock loop is stabilized by a second interlocking feedback loop composed of retinoic acid receptor-related

**eLife digest** Like many other animals, our behavior often follows a familiar pattern each day. We tend to wake up with the morning sunlight and start by eating breakfast to satisfy our hunger. Then, at night, most of us sleep, and our bodies use chemical building blocks from the day's meals to replenish and repair our tissues. As such, its not just our behavior that shows a daily cycle. The countless chemical reactions that keep us alive, collectively referred to as our metabolism, also change over the course of each day.

Daily patterns of activity, known as circadian rhythms, can be seen even at the level of individual cells. Each cell in the body has its own molecular clock that works by rhythmically cycling the levels of different molecules. Proteins called CLOCK and BMAL1 trigger the production of proteins PER and CRY. As levels of PER and CRY rise, they interfere with CLOCK and BMAL1, essentially switching off their own production. Then, levels of PER and CRY fall and the cycle starts again. Thus, these molecules rise and fall throughout the day to drive circadian rhythms, similar to how a pendulum swings back and forth to keep a clock ticking.

Metabolism and circadian rhythms are clearly linked, but it is not well understood how this works. Now, Wible, Ramanathan et al. identify a protein called NRF2 as an important bridge between the molecular clock and metabolism. NRF2 is a activated by hydrogen peroxide, a byproduct of cell metabolism, and when activated this protein grabs hold of DNA to increase the activity of specific genes.

A combination of experiments revealed that mouse liver cells need NRF2 to maintain a normal circadian rhythm, and a closer inspection of the liver cells revealed that NRF2 specifically attaches to part of the gene for a clock protein called CRY2. This enhances the production of this protein, which in turn, switches CLOCK and BMAL1 off. In this way, NRF2 links metabolism signals to the ticking of the circadian clock.

Previous research has shown a link between shift work, which disrupt these rhythms, and metabolic diseases like obesity and diabetes. Understanding more about the underlying molecular biology that links metabolism to circadian rhythms could help scientists to find new ways to improve public health.

DOI: https://doi.org/10.7554/eLife.31656.002

orphan receptor (RORα/β/γ)-dependent transcription of *Bmal1*, a process inhibited by the *Bmal1* downstream targets Rev-Erbα/β (*Nr1d1/Nr1d2*) (*Preitner et al., 2002*; *Sato et al., 2004*; *Liu et al., 2008*; *Cho et al., 2012*). In addition to the regulation of known clock genes, the CLOCK/BMAL1-complex also regulates the expression of thousands of other genes, which drive rhythmic changes in various tissue-specific physiological and cellular processes (*Zhang et al., 2014*; *Perelis et al., 2015*).

Many clock controlled output genes encode metabolic enzymes leading to diurnal oscillations in carbohydrate flux (*Asher and Schibler, 2011*; *Bass, 2012*; *Zhang and Kay, 2010*; *Bass and Takahashi, 2010*). Rhythmic carbohydrate catabolism and mitochondrial oxidative phosphorylation, particularly in hepatocytes (*Jacobi et al., 2015*; *Peek et al., 2013*), likely contribute to oscillations in reactive oxygen species (ROS) (*Pekovic-Vaughan et al., 2014*; *Stangherlin and Reddy, 2013*; *Milev and Reddy, 2015*). Signaling through oxidative intermediates (*Sies, 2014*) modulates an array of redox sensitive transcription factors and enzymes to regulate gene expression, metabolic flux, and timekeeping (*Pekovic-Vaughan et al., 2014*; *Xu et al., 2012*; *Wible and Sutter, 2017*). Appropriate timing and magnitude of ROS signaling likely facilitates the temporal segregation and transition between incompatible metabolic processes and increases overall metabolic efficiency (*Milev and Reddy, 2015*; *Jacobi et al., 2015*).

A key transcription factor, regulated by oxidative signaling, controlling both metabolic (*Mitsuishi et al., 2012*) and circadian (*Rey et al., 2016*; *Yang et al., 2014*) gene expression is NF-E2 related-factor 2 (NRF2). The activity of NRF2 is antagonized by the cytoplasmic repressor kelch like ECH associated protein 1 (KEAP1). KEAP1 destabilizes NRF2 by facilitating its ubiquitinylation and eventual proteasomal degradation. Activators of NRF2 signaling relieve NRF2 from KEAP1-mediated repression through the oxidation or covalent modification of KEAP1 cysteine residues. Well-studied chemical NRF2 activators include the clinically relevant electrophiles 3H-1,2-dithiole-3-thione (D3T)

and 1-[2-cyano-3,12-dioxooleana-1,9 (11)-dien-28-oyl]imidazole (CDDO-Im), and the oxidants *tert*-butylhydroquinone (tBHQ) (*Imhoff and Hansen, 2010*) and hydrogen peroxide ($H_2O_2$). Modification of KEAP1 cysteine residues by these agents results in an increased NRF2 half-life that facilitates nuclear translocation and accumulation (*Wible and Sutter, 2017*; *Suzuki and Yamamoto, 2015*; *Kwak et al., 2004*; *Saito et al., 2016*). The expression of *Nrf2* is under the transcriptional regulation of the CLOCK/BMAL1-complex (*Pekovic-Vaughan et al., 2014*; *Xu et al., 2012*; *Zhang et al., 2009*; *Lee et al., 2013*). CLOCK/BMAL1-dependent *Nrf2* regulation gives rise to diurnal patterns in NRF2 signaling, which underlies the rhythmic expression of antioxidant and metabolic enzymes as well as NADPH reducing equivalents and glutathione biosynthesis (*Xu et al., 2012*). Here, we report the effects of NRF2 gain- and loss-of-function on circadian gene expression and rhythmicity, elaborating the coupling of NRF2 and clock and the role of NRF2 to integrate cellular redox status into timekeeping.

## Results

### Keap1/Nrf2-activators affect circadian gene expression in mouse liver and perturb circadian rhythmicity in vitro

Microarray data (GSE99199; *Wible et al., 2018*) characterizing global gene expression changes in mouse liver in response to D3T suggested a possible alteration in circadian gene expression in response to chemical NRF2-activation (*Figure 1—figure supplement 1*). To further characterize the effect of D3T on circadian gene expression and to determine whether those effects were dependent on NRF2, we measured the RNA expression of several circadian genes in the liver of Wt and *Nrf2-/-* mice treated with three doses of 300 μmol/kg bodyweight D3T (every other day, with sampling 24 hr following the third dose) (*Figure 1A*). While E-box and D-box-containing clock genes (*Nr1d1*, *Nr1d2*, *Dbp*, T*ef*, and *Per3*) were up-regulated in response to D3T in Wt mice, genes that are regulated primarily via RAR-related orphan receptor response elements (ROREs) (*Bmal1*, *Npas2*, *E4bp4*) were down-regulated, consistent with the network features of the molecular clock (*Ueda et al., 2005*; *Liu et al., 2008*; *Baggs et al., 2009*). The full induction of E-box- and D-box-containing genes (*Per3*, *Nr1d1*, *Nr1d2*, *Dbp*, and *Tef*) in response to D3T-treatment required the presence of the KEAP1/NRF2 signaling pathway, as their induction was significantly compromised in *Nrf2-/-* mice. *Nrf2* deficiency did not alter the down-regulation of *Bmal1*, *Cry1*, or *E4bp4*, suggesting their independence of NRF2 and that the circadian network effect did not transmit to RORE-mediated transcription.

Next, we evaluated whether the effects of D3T on circadian gene expression observed in vivo manifested in alterations in circadian function in vitro and whether those potential alterations were dependent on NRF2. Chemical activation of NRF2 occurs through either electrophilic or oxidative modification of KEAP1, antagonizing its repression of NRF2. To assess the effect of each type of KEAP1 modification and subsequent NRF2 activation on the circadian rhythm, we treated Wt and *Nrf2-/-* mouse embryonic fibroblast (MEF) cell lines harboring a Per2 promoter-driven luciferase reporter (Per2:Luc) with a single dose of 100 μM D3T (*Figure 1B*) or 50 μM tBHQ (*Figure 1C*). Both treatments resulted in significantly reduced rhythm amplitude, an effect that was not observed in the *Nrf2-/-* cells. As a primary regulator of the antioxidant response pathway, it is likely that an endogenous NRF2 activating signal is $H_2O_2$, generated as a byproduct of metabolic flux. To evaluate the potential effect on circadian rhythmicity as a result of the activation of NRF2 by an endogenous oxidative signal, we treated MEFs at the beginning of the cycle with a low concentration (100 μM) of $H_2O_2$ (*Figure 1D*). Similar to both D3T and tBHQ treatment, we observed significant NRF2-dependent reductions in rhythm amplitude in the presence of $H_2O_2$.

As a well-characterized, clinically relevant, NRF2-activating compound, we also evaluated the effect of CDDO-Im on circadian rhythmicity (*Figure 1—figure supplement 2*). Treatment with CDDO-Im caused significant reductions in both amplitude and period length (Treatment), that were noticeably absent in *Nrf2-/-* MEFs. Because poor circadian oscillations in the presence of CDDO-Im may be a product of chemical toxicity, we replaced the treatment medium in cell lines of both genotypes and continued monitoring rhythmic bioluminescence patterns (Recovery). The restoration of normal circadian rhythmicity in the treated Wt cells served as a confirmation of good cell health and a dependency on CDDO-Im for the alterations in the circadian parameters observed.

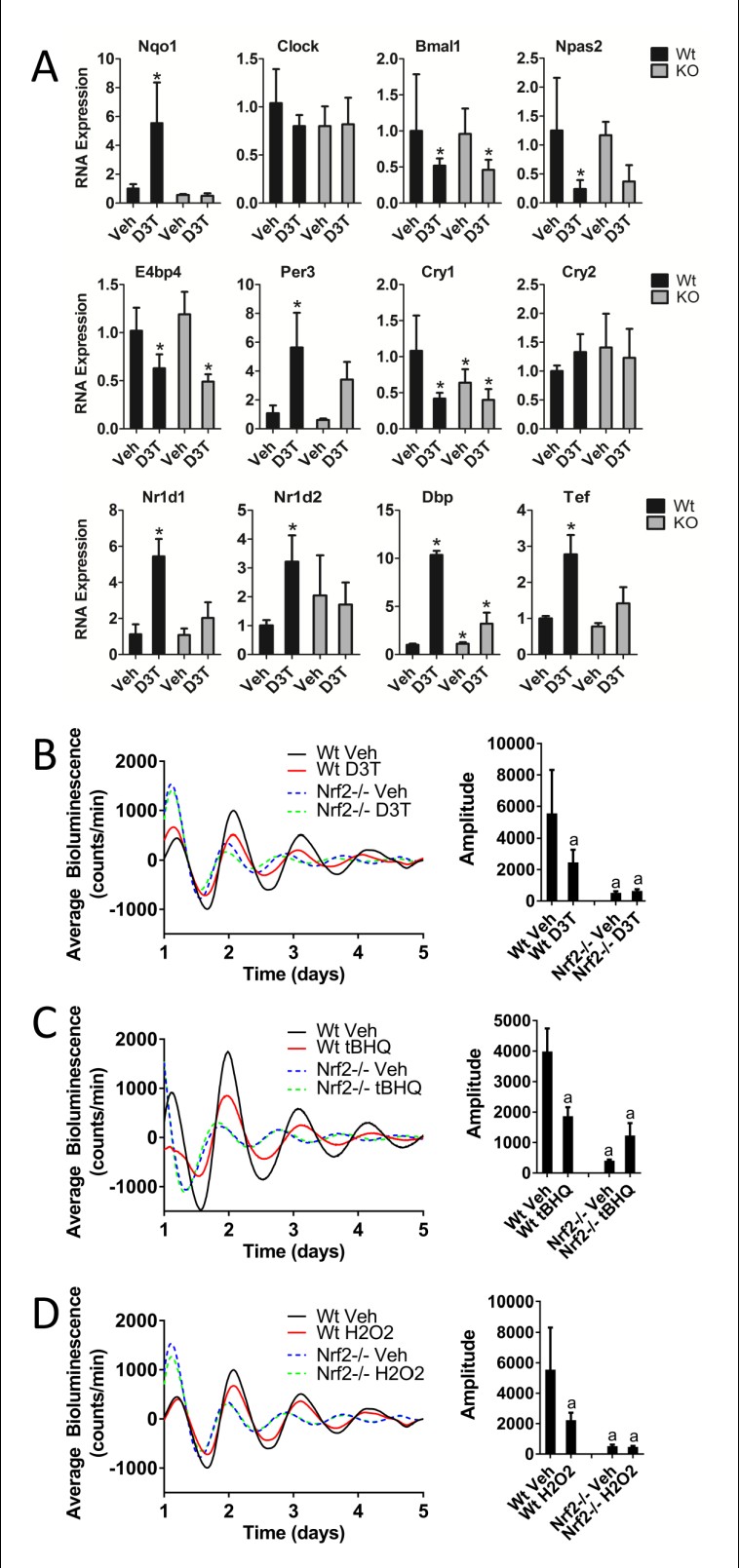

**Figure 1.** Keap1/Nrf2-activators affect circadian gene expression in mouse liver tissue and perturb circadian rhythmicity in vitro. (**A**) Expression of circadian genes in the liver of Wt and *Nrf2-/-* (KO) mice treated with either vehicle (Veh) or 300 μmol/kg bw D3T. All expression values were determined by qPCR and normalized to *Gapdh*. Data is shown as the average fold-change ±standard deviation (n = 4 animals/treatment/genotype each) relative to

*Figure 1 continued*

the expression in the Wt Veh sample, which was set to 1. Results were analyzed using Holm-Sidak's multiple comparisons test; * indicates a p-value<0.05 relative to the Wt Veh control. (**B–D**) Per2:Luc-driven bioluminescence in Wt or *Nrf2-/-* MEFs in the presence of DMSO (0.05%) (Veh), (**B**) 100 µM D3T, (**C**) 50 µM tBHQ, or (**D**) 100 µM H$_2$O$_2$. Average luminescence recordings are shown on the left in each panel. Amplitude is expressed as an average ±standard deviation (n = 3). Results were analyzed using a two-way ANOVA followed by Tukey's multiple comparisons test; a indicates a p-value<0.05 relative to the Wt Veh control.

DOI: https://doi.org/10.7554/eLife.31656.003

The following figure supplements are available for figure 1:

**Figure supplement 1.** Circadian pathway enriched by D3T in mouse liver.
DOI: https://doi.org/10.7554/eLife.31656.004
**Figure supplement 2.** Effect of CDDO-Im on circadian rhythmicity in MEFs.
DOI: https://doi.org/10.7554/eLife.31656.005

## Nrf2 is required for normal circadian timekeeping

In our earlier experiments, we observed inherently low bioluminescence rhythm amplitudes (*Figure 1B–D*) in *Nrf2-/-* MEFs, indicating that NRF2 may be a required component of the circadian clockwork. To test this hypothesis, we determined rhythmic bioluminescence patterns in Wt and *Nrf2-/-* MEFs. In addition to the previously observed reduction in rhythm amplitude, the loss of *Nrf2* also resulted in a significant decrease in period length (*Figure 2A*, *Figure 2—figure supplement 1*). To confirm that the circadian phenotype observed in these cells was a product of the loss of NRF2 signaling and not an artifact of the established cell lines, we genetically reconstituted *Nrf2* expression in the *Nrf2-/-* MEF cells (*Figure 2B*, *Figure 2—figure supplement 2*). The restoration of *Nrf2* expression rescued both circadian amplitude and period length relative to the parental *Nrf2-/-* cell line, but fell short of achieving Wt rhythmicity. As a marker of NRF2 activity, we measured increased *Nqo1* expression, confirming that exogenous *Nrf2* expression yields functional NRF2 protein (*Figure 2C*). Consistent with *Nrf2* expression being driven by a constitutive CMV promoter, NRF2 protein expression in the rescued cell line was constitutively elevated at 24 hr and 36 hr post-synchronization. This contrasted with circadian NRF2 expression observed in Wt MEFs (*Figure 2D*, *Figure 2—figure supplement 3*).

To independently validate the circadian phenotype observed in *Nrf2-/-* MEFs, we established a shRNA-mediated *Nrf2* knockdown MEF cell line. *Nrf2* knockdown led to significant reductions in both rhythm amplitude and period length (*Figure 2E*), albeit to a lesser extent than what was observed in *Nrf2-/-* MEFs. The efficiency of *Nrf2* knockdown was confirmed by measurements of significantly reduced NRF2 protein abundance (*Figure 2F*). *Nrf2* and *Nqo1* RNA expression was also significantly down-regulated as a result of shNrf2 transduction (*Figure 2G*). Interestingly, *Nr1d1* expression was also decreased in response to shNrf2, consistent with previous reports (*Rey et al., 2016*; *Yang et al., 2014*) and our results from mouse liver (*Figure 1A*), suggesting that *Nr1d1* may be transcriptionally regulated by NRF2 (*Figure 2G*). Together, these data support the idea that the disrupted circadian rhythmicity observed in *Nrf2-/-* MEFs is a direct effect of the loss of *Nrf2* and implicate a role for NRF2 in the regulation of rhythm amplitude and period length.

Reduced *Nr1d1* expression in shNrf2 cell lines suggested that the mechanistic input of NRF2 into the clockwork may be through the regulation of *Nr1d1* expression. We determined the temporal abundance patterns of NR1D1 protein as a function of the loss of *Nrf2* in an attempt to gain insight as to whether changes in *Nr1d1* gene expression had an effect on a rhythmic output. In *Nrf2-/-* MEFs, the peak of NR1D1 protein accumulation was delayed by 4 hr relative to the Wt (*Figure 2H*). Delayed NR1D1 expression, as opposed to the complete loss, in *Nrf2-/-* cells suggests that *Nr1d1* is likely regulated by additional transcription factors beyond NRF2. The improper temporal regulation of NR1D1 in the absence of *Nrf2*, however, may contribute to the perturbed circadian rhythmicity observed in MEFs lacking *Nrf2*.

## Genetic activation of NRF2 perturbs circadian rhythmicity and regulates *Nr1d1* gene expression

Previous reports regarding the cross-talk between NRF2 and the circadian rhythm have demonstrated that activation of NRF2 in the presence of 6-aminonicotinamide (*Rey et al., 2016*) or

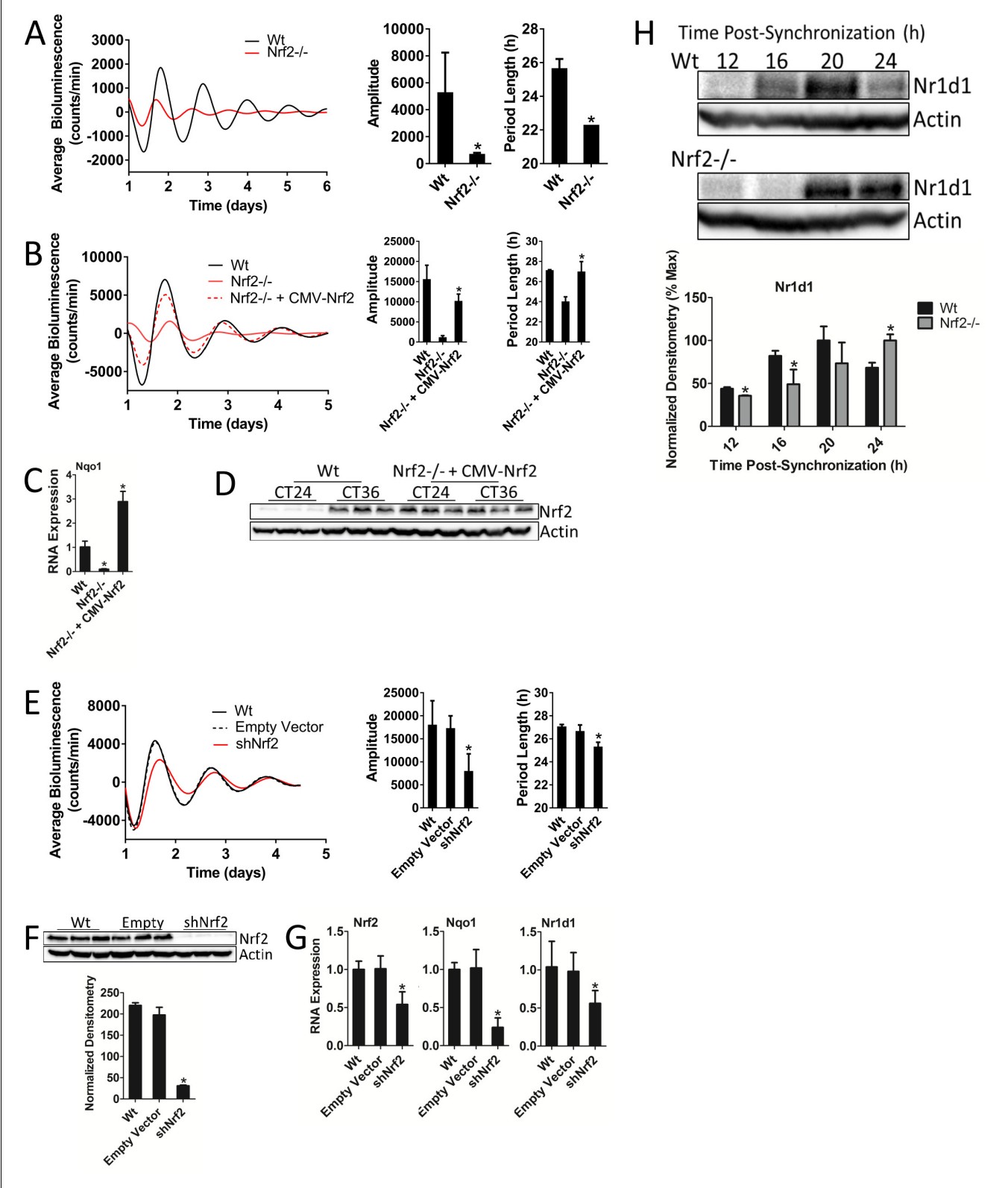

**Figure 2.** Nrf2 is required for normal circadian timekeeping. (**A**) Per2:Luc-driven bioluminescence from Wt and *Nrf2-/-* MEFs. Average luminescence recordings are shown on the left. Amplitude and period length are expressed as an average ±standard deviation (n = 3). Results were analyzed using Student's *t*-test; * indicates a p-value<0.05 relative to Wt. (**B**) Per2:Luc-driven bioluminescence from Wt, *Nrf2-/-*, and *Nrf2-/-* MEFs stably expressing an *Nrf2* expression construct (Nrf2-/- + CMV-Nrf2). Average luminescence recordings are shown on the left. Amplitude and period length are expressed as

*Figure 2 continued on next page*

*Figure 2 continued*

an average ±standard deviation (n = 3). Results were analyzed using Student's *t*-test; * indicates a p-value<0.05 relative to *Nrf2-/-*. (C) *Nqo1* gene expression in Wt, *Nrf2-/-*, and *Nrf2-/-* + CMV-Nrf2 MEFs. Expression values were determined by qPCR and normalized to *Gapdh*. Data is shown as the average fold-change ±standard deviation (n = 3) relative to the expression in Wt, which was set to 1. Results were analyzed using a one-way ANOVA followed by Dunnett's multiple comparisons test; * indicates a p-value<0.05 relative to Wt. (D) NRF2 expression in 50 μg of whole cell lysate from Wt and *Nrf2-/-* + CMV-Nrf2 MEFs and harvested 24 hr or 36 hr post-synchronization. β-Actin, which was unchanged by exogenous Nrf2 expression, was used as a loading control. Average densitometric values ± standard deviation equal 14.22 ± 2.39 and 107.14 ± 16.68 for Wt 24 hr and 36 hr post-synchronization, respectively. Average densitometric values (±standard deviation) equal 157.4 ± 12.37 and 160.16 ± 18.09 for *Nrf2-/-* + CMV-Nrf2 24 hr and 36 hr post-synchronization, respectively. (E) Per2:Luc-driven bioluminescence from Wt and Wt MEFs stably expressing an empty shRNA vector (Empty Vector) or a shNrf2 vector (shNrf2). Average luminescence recordings are shown on the left. Amplitude and period length are expressed as an average ±standard deviation (n = 4). Results were analyzed using Student's *t*-test; * indicates a p-value<0.05 relative to the Empty Vector. (F) NRF2 expression in 50 μg of whole cell lysate from Wt, Empty Vector (Empty), and shNrf2 MEFs. β-Actin, which was unchanged by shRNA expression, was used as a loading control. Values are expressed as the normalized average densitometry ±standard deviation (n = 3). Results were analyzed using Student's t-test; * indicates a p-value<0.05 relative to the Empty Vector. (G) *Nrf2*, *Nqo1*, and *Nr1d1* gene expression in Wt, Empty Vector, and shNrf2 MEFs. Expression values were determined by qPCR and normalized to *Gapdh*. Data is shown as the average fold-change ±standard deviation (n = 3) relative to the expression in Wt, which was set to 1. Results were analyzed using Student's *t*-test; * indicates a p-value<0.05 relative to the Empty Vector. (H) NR1D1 protein in 50 μg of whole cell lysate from Wt and *Nrf2-/-* MEFs at the indicated times post-synchronization. β-Actin, which was unchanged by genotype or collection time, was used as a loading control. Values are expressed as the normalized average densitometry ±standard deviation (n = 3) relative to the time point in which expression was maximal, which was set to 100%. Results were analyzed using Student's t-test; * indicates a p-value<0.05 relative to the time matched Wt sample.

DOI: https://doi.org/10.7554/eLife.31656.006

The following figure supplements are available for figure 2:

**Figure supplement 1.** Nrf2 is required for normal circadian timekeeping.
DOI: https://doi.org/10.7554/eLife.31656.007
**Figure supplement 2.** Nrf2 is required for normal circadian timekeeping.
DOI: https://doi.org/10.7554/eLife.31656.008
**Figure supplement 3.** Characterization of Nrf2 circadian expression dynamics for *Figure 4D*.
DOI: https://doi.org/10.7554/eLife.31656.009

hyperoxic conditions (*Yang et al., 2014*) lead to the transcriptional regulation of *Nr1d1*. We are currently unaware of any studies evaluating the effect of genetic NRF2 gain-of-function on circadian gene transcription or the circadian rhythm. Because genetic gain-of-function and chemical activation have been shown to lead to distinct NRF2-dependent gene responses (*Yates et al., 2009*), we explored the effects of genetic NRF2 activation on circadian rhythmicity and gene expression.

We established *Keap1-/-* MEFs harboring a Per2:Luc reporter to determine the circadian phenotype as a function of genetic NRF2 activation. Constitutive NRF2 activation in the *Keap1-/-* background resulted in significant reductions in both amplitude and period length (*Figure 3A*).

To confirm our observations in the *Keap1-/-* MEFs, we overexpressed *Nrf2* in Wt MEFs by transducing these cells with a CMV-driven *Nrf2* expression construct. Overexpression of NRF2 in a Wt background resulted in reduced circadian amplitude and period length in manner similar to the *Keap1-/-* cell line (*Figure 3B*). The overexpression and enhanced transcriptional activity of NRF2 in this cell line, were validated by measurement of elevated NRF2 protein (*Figure 3C*) and NQO1 protein and RNA expression (*Figure 3C–D*) in the Wt +CMV-Nrf2 cells. Moreover, we observed elevated levels of NR1D1 protein (*Figure 3C*) and RNA (*Figure 3D*) in response to *Nrf2* overexpression, consistent with NRF2 contributing to *Nr1d1* gene expression. Interestingly, the circadian phenotypes resulting from the overexpression of NRF2 recapitulated the phenotype of genetically rescued *Nrf2-/-* MEFs, suggesting that the timing or absolute stoichiometry of NRF2 protein abundance are likely to be important for proper circadian function.

In addition to genetic gain-of-function, the activation of NRF2 by D3T also had an effect on *Nr1d1* expression. In response to 100 μM D3T, the expression of both *Nqo1* and *Nr1d1* were significantly induced (*Figure 3E*). These effects were noticeably absent in D3T-treated *Nrf2-/-* MEFs. We did observe a slight, but significant, induction of *Nr1d1* expression in *Nrf2-/-* MEFs in the absence of D3T-treatment, consistent with the previously reported induction of *Nr1d1* under oxidative intracellular conditions (*Yang et al., 2014*), a condition likely to be present in *Nrf2-/-* MEFs.

Using a position weight matrix implemented in the JASPAR CORE database (*Mathelier et al., 2016*), we identified putative antioxidant response elements (AREs) in the regions surrounding the

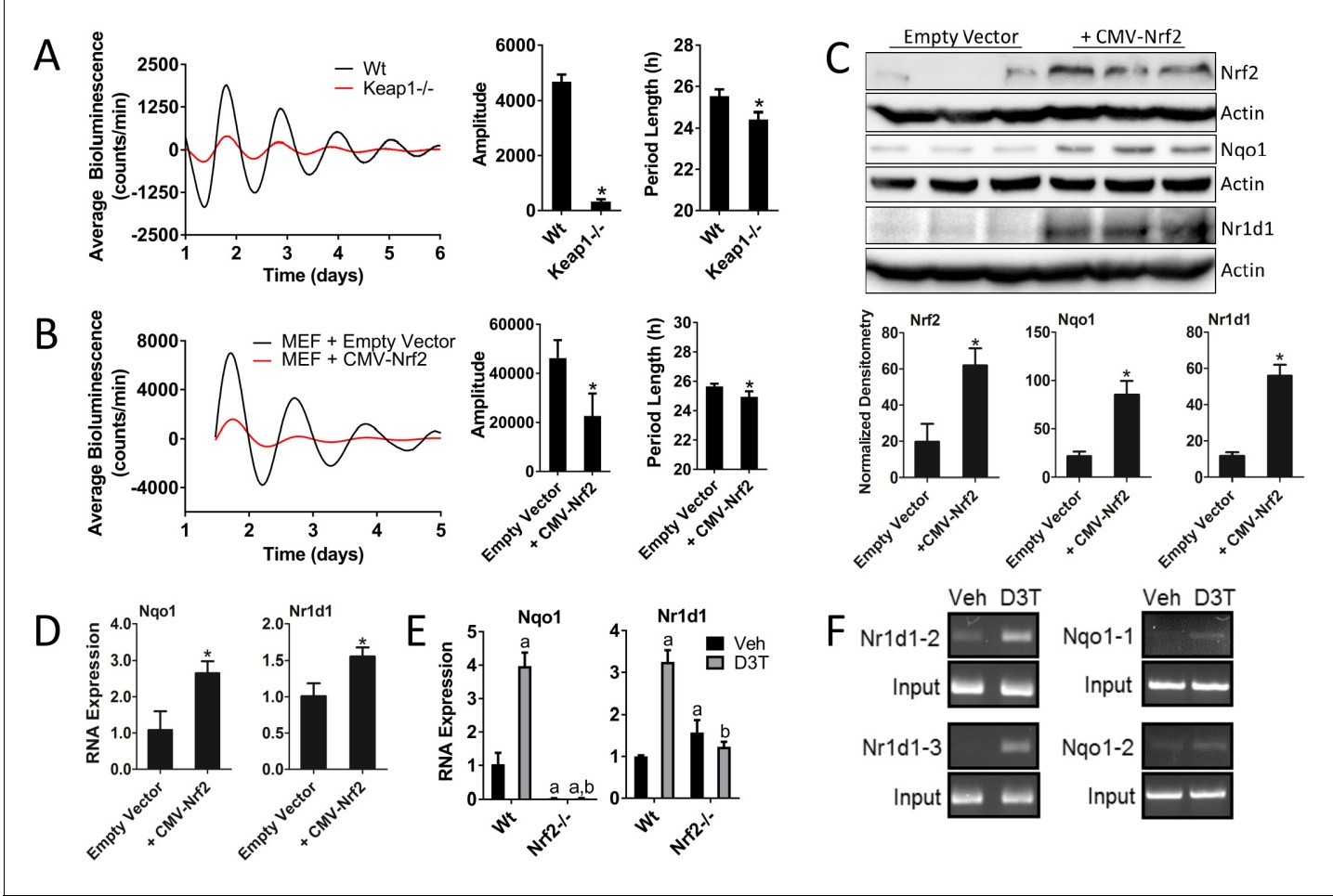

**Figure 3.** Genetic activation of NRF2 perturbs circadian rhythmicity and regulates *Nr1d1* gene expression. (**A**) Per2:Luc-driven bioluminescence from Wt and *Keap1*-/- MEFs. Average luminescence recordings are shown on the left. Amplitude and period length are expressed as an average ±standard deviation (n = 5). Results were analyzed using Student's *t*-test; * indicates a p-value<0.05 relative to Wt. (**B**) Per2:Luc-driven bioluminescence from Wt MEFs stably expressing an empty vector (Wt) or a *Nrf2* expression construct (CMV-Nrf2). Average luminescence recordings are shown on the left. Amplitude and period length are expressed as an average ±standard deviation (n = 3). Results were analyzed using Student's *t*-test; * indicates a p-value<0.05 relative to Wt. (**C**) NRF2, NQO1, and NR1D1 protein in 50 µg of whole cell lysate from Wt and Wt +CMV-Nrf2 MEFs. β-Actin, which was unchanged by exogenous *Nrf2* expression, was used as a loading control. Values are expressed as the normalized average densitometry ±standard deviation (n = 3). Results were analyzed using Student's *t*-test; * indicates a p-value<0.05 relative to the Wt. (**D**) *Nqo1* and *Nr1d1* gene expression, in Wt and Wt +CMV-Nrf2 MEFs. Expression values were determined by qPCR and normalized to *Gapdh*. Data is shown as the average fold-change ±standard deviation (n = 3) relative to the expression in Wt, which was set to 1. Results were analyzed using Student's *t*-test; * indicates a p-value<0.05 relative to the Wt. (**E**) *Nqo1* and *Nr1d1* gene expression in Wt and *Nrf2*-/- MEFs treated with DMSO (Veh) or 100 µM D3T for 24 hr. Expression values were determined by qPCR and normalized to *Gapdh*. Data is shown as the average fold-change ±standard deviation (n = 3) relative to the expression in the Wt Veh, which was set to 1. Results were analyzed using a two-way ANOVA followed by Tukey's multiple comparisons test; a indicates a p-value<0.05 relative to the Wt Veh control, b indicates a p-value<0.05 relative to treated Wt sample. (**F**) Chromatin immunoprecipitation-PCR using an NRF2 antibody to detect binding of NRF2 to the indicated gene promoters. Wt MEFs were treated with DMSO (Veh) or 100 µM D3T for 24 hr. Putative ARE sequences can be found in *Supplementary file 2* and their location in the respective promoter regions can be found in *Figure 3—figure supplement 1*.

DOI: https://doi.org/10.7554/eLife.31656.010

The following figure supplement is available for figure 3:

**Figure supplement 1.** Genetic activation of NRF2 perturbs circadian rhythmicity and regulates *Nr1d1* gene expression.
DOI: https://doi.org/10.7554/eLife.31656.011

promoters of *Nr1d1* and *Nqo1* (*Figure 3—figure supplement 1*, *Supplementary files 1–2*). Chromatin immunoprecipitation of NRF2 in MEFs treated with 100 µM D3T for 24 hr demonstrated direct binding of NRF2 to two of the three ARE elements identified in the *Nr1d1* promoter, namely Nr1d1-2 and Nr1d1-3, and two ARE elements in *Nqo1*, namely Nqo1-1 and Nqo1-2 (*Figure 3F*). The binding of NRF2 to the elements Nr1d1-3 and Nqo1-1 are novel findings, while the binding of NRF2 to the element in Nr1d1-2 (*Yang et al., 2014*) and Nqo1-1 (*Nioi et al., 2003*) confirm previous reports. Together these data demonstrate that genetic or pharmacological activation of NRF2 impinges on circadian clock function, which correlates with enhanced occupation of NRF2 on the *Nr1d1* promoter and an induction of *Nr1d1* expression.

## Nrf2 couples the intracellular redox state to the molecular circadian clock, reinforces rhythm amplitude, and shifts circadian phase

As the key regulator of an antioxidant stress response, we hypothesized that the activation of NRF2 by appropriately timed endogenous levels of oxidants would cause a reinforcement of the circadian amplitude, similar to what has been observed in plants (*Zhou et al., 2015*). To determine the timing of endogenous oxidative signals relative to the circadian-mediated expression of NRF2, we measured the oxidation state of peroxiredoxin (PRDX-SO$_{2/3}$) as a biomarker of an oxidized intracellular environment (*Edgar et al., 2012*) in synchronized MMH-D3 hepatocytes (*Figure 4A*). Levels of hyperoxidized PRDX were maximal 28 hr post-synchronization (red arrow), which preceded peak NRF2 protein accumulation (black arrow) by 12 hr. This 12 hr lag phase is likely a product of the kinetics of NRF2 activation. NRF2-activating signals inhibit KEAP1-mediated NRF2 turnover, resulting in KEAP1 becoming saturated with bound NRF2. Due to the short NRF2 half-life, a lag time would be expected to allow for the saturation of KEAP1 NRF2-binding sites and the accumulation of newly synthesized active NRF2 following the activation of the pathway (*Wible and Sutter, 2017*). The timing of oxidative signaling (*Figure 4A*), however, is consistent with what has been previously observed in mouse liver (*Xu et al., 2012*; *Edgar et al., 2012*), indicating that the generation of endogenous oxidative signals occurs at a similar time in vitro relative to *Nrf2* expression as it does in vivo.

To test our hypothesis that appropriately timed oxidative NRF2 activation could lead to circadian amplitude reinforcement, we treated MEFs with 100 µM H$_2$O$_2$, a concentration likely to be physiologically relevant, at a circadian time corresponding to the endogenous peak of oxidative signaling determined in hepatocytes (*Figure 4B*, *Figure 4—figure supplement 1*). Activation of NRF2 in response to an oxidative signal at this time resulted in significantly increased amplitude, without affecting period length. A similar effect was also observed in the presence of another NRF2-activating compound D3T (*Figure 4—figure supplement 2*), suggesting that amplitude reinforcement is a function of NRF2 activity and not a side-effect of any one particular chemical. In addition to reinforcing circadian amplitude, activation of NRF2 by H$_2$O$_2$ at this time also resulted in a subtle advance of the circadian phase, which became more prominent at higher concentrations of H$_2$O$_2$ (500 µM) (*Figure 4—figure supplement 3*).

To confirm that the reinforcement of circadian amplitude in response to oxidative signaling was dependent on NRF2 activation, we treated Wt and *Nrf2-/-* MEFs with 100 µM H$_2$O$_2$ at a time consistent with the generation of endogenous oxidative signals. Treatment with H$_2$O$_2$, again, resulted in circadian amplitude reinforcement, which was absent in H$_2$O$_2$-treated *Nrf2-/-* MEFs (*Figure 4C*, *Figure 4—figure supplement 4*). Reinforcement of circadian amplitude, following the addition of H$_2$O$_2$, could be blocked in the presence of an antioxidant (N-acetylcysteine; NAC) (*Figure 4D*), suggesting that oxidation was indeed the relevant signal responsible for the reinforcement of circadian amplitude. Interestingly, treatment with NAC alone resulted in significantly decreased rhythm amplitude and period length, suggesting that endogenous oxidative signals (*Sies, 2017*) may play a role in the maintenance of circadian timekeeping.

Similar dependence on NRF2 for amplitude reinforcement in response to H$_2$O$_2$ was also observed when the treatment was applied at a different time in the circadian cycle (*Figure 4E*). In contrast to what we observed when applying H$_2$O$_2$ prior to the peak of PER2 expression, addition of H$_2$O$_2$ prior to the trough resulted in a subtle delay of the circadian phase. Interestingly, when H$_2$O$_2$ was added to the cultures at the nadir of PER2 expression, the circadian phase appeared to be delayed or possibly reset (*Figure 4F*, *Figure 4—figure supplement 5*). This effect was not observed in *Nrf2-/-* MEFs, indicating that H$_2$O$_2$-induced phase-dependent shifts in circadian phase may occur through

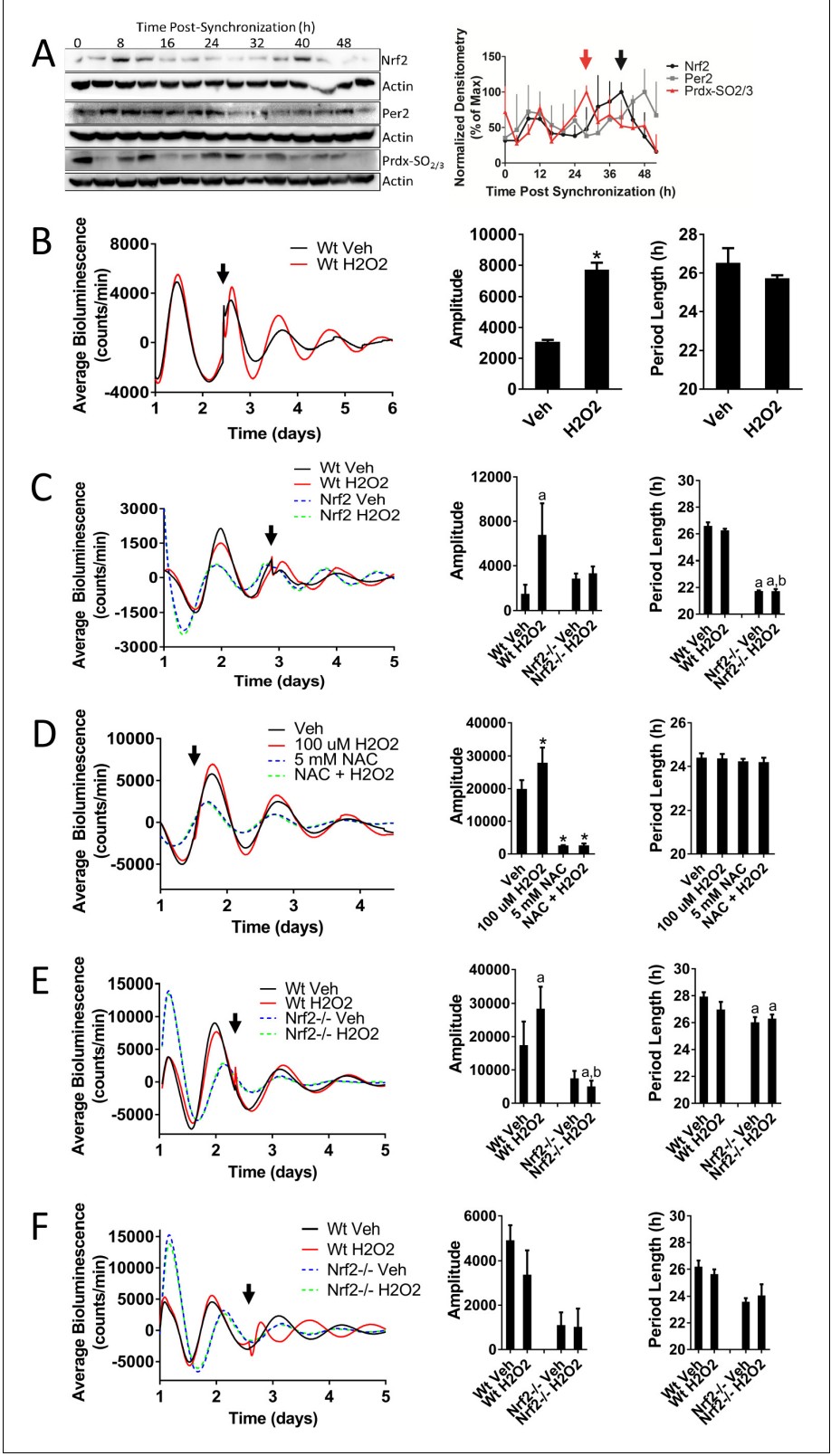

**Figure 4.** Nrf2 couples the intracellular redox state to the molecular circadian clock, reinforces rhythm amplitude, and shifts circadian phase. (**A**) NRF2, PER2, and PRDX-SO$_{2/3}$ protein in 50 μg of whole cell lysate from differentiated MMH-D3 hepatocytes at the indicated times post-synchronization. β-Actin, which was unchanged across time points, was used as a loading control. Values are expressed as the normalized average

*Figure 4 continued on next page*

*Figure 4 continued*

densitometry ±standard error mean (n = 3) relative to the time point in which expression was maximal, which was set to 100%. Red and black arrows indicate the time of peak expression of PRDX-SO$_{2/3}$ and NRF2, respectively. (B) Per2:Luc-driven bioluminescence from Wt MEFs treated with DMSO (0.05%) (Veh) or 100 μM H$_2$O$_2$ at the time indicated by the arrow. Average luminescence recordings are shown on the left. Amplitude and period length are expressed as an average ±standard deviation (n = 3). Results were analyzed using Student's $t$-test; * indicates a p-value<0.05 relative to Veh. (C) Per2:Luc-driven bioluminescence from Wt and *Nrf2-/-* MEFs treated with water (Veh) or 100 μM H$_2$O$_2$ at the time indicated by the arrow. Average luminescence recordings are shown on the left. Amplitude and period length are expressed as an average ±standard deviation (n = 3). Results were analyzed using two-way ANOVA followed by Tukey's multiple comparisons test; a indicates a p-value<0.05 relative to the Wt Veh control, b indicates a p-value<0.05 relative to the H$_2$O$_2$-treated Wt sample. (D) Per2:Luc-driven bioluminescence from Wt MEFs in the presence of water (Veh) or 5 mM NAC then treated with 100 μM H$_2$O$_2$ at the time indicated by the arrow. Average luminescence recordings are shown on the left. Amplitude and period length are expressed as an average ±standard deviation (n = 3). Results were analyzed using one-way ANOVA followed by Dunnett's multiple comparisons test; * indicates a p-value<0.05 relative to the Veh control. (E–F) Per2:Luc-driven bioluminescence from Wt and *Nrf2-/-* MEFs treated with water (Veh) or 100 μM H$_2$O$_2$ at the time indicated by the arrow. Average luminescence recordings are shown on the left in each panel. Amplitude and period length are expressed as an average ±standard deviation (n = 3). Results were analyzed using two-way ANOVA followed by Tukey's multiple comparisons test; a indicates a p-value<0.05 relative to the Wt Veh control, b indicates a p-value<0.05 relative to the H$_2$O$_2$-treated Wt sample.
DOI: https://doi.org/10.7554/eLife.31656.012

The following figure supplements are available for figure 4:

**Figure supplement 1.** Nrf2 couples the intracellular redox state to the molecular circadian clock, reinforces rhythm amplitude, and shifts circadian phase.
DOI: https://doi.org/10.7554/eLife.31656.013

**Figure supplement 2.** Nrf2 couples the intracellular redox state to the molecular circadian clock, reinforces rhythm amplitude, and shifts circadian phase.
DOI: https://doi.org/10.7554/eLife.31656.014

**Figure supplement 3.** Nrf2 couples the intracellular redox state to the molecular circadian clock, reinforces rhythm amplitude, and shifts circadian phase.
DOI: https://doi.org/10.7554/eLife.31656.015

**Figure supplement 4.** Nrf2 couples the intracellular redox state to the molecular circadian clock, reinforces rhythm amplitude, and shifts circadian phase.
DOI: https://doi.org/10.7554/eLife.31656.016

**Figure supplement 5.** Nrf2 couples the intracellular redox state to the molecular circadian clock, reinforces rhythm amplitude, and shifts circadian phase.
DOI: https://doi.org/10.7554/eLife.31656.017

an NRF2-dependent mechanism. While shifts in circadian phase in response to oxidative signals have been observed previously across multiple cell lines (*Tahara et al., 2016*; *Putker et al., 2018*), our data implicates the activation of NRF2 as the likely mechanism through which oxidants signal into the clockwork. NRF2-dependent input into the clock effectively couples redox homeostasis and timekeeping yielding a molecular framework to enhance oscillator robustness and shift circadian phase.

## Nrf2 is required to maintain mouse hepatocyte and liver circadian pace

Similar to our observations in MEFs, shRNA-mediated *Nrf2* knockdown in hepatocytes caused significant amplitude and period length reductions (*Figure 5A*, *Figure 5—figure supplement 1*). Hepatocytes harboring shNrf2 showed a 50% reduction in total NRF2 protein expression and a greater than 90% reduction in NQO1 (*Figure 5B*). These data indicate that a reduction in total NRF2 protein resulted in markedly lower transcriptional activity, which paralleled significant reductions in rhythm amplitude and period length.

To evaluate the coupling of NRF2 to the clock in vivo, we characterized the rhythmic bioluminescence patterns in liver and lung organotypic slices from two different *Nrf2*-null PER2::LUC mouse strains (MYM and YWK). Loss of *Nrf2* in the liver of mice from both strains resulted in a significant alteration in circadian period length (*Figure 5C and E*). Surprisingly, we observed no deleterious effects of *Nrf2* loss on circadian rhythmicity in the lung tissue of either strain (*Figure 5D and F*),

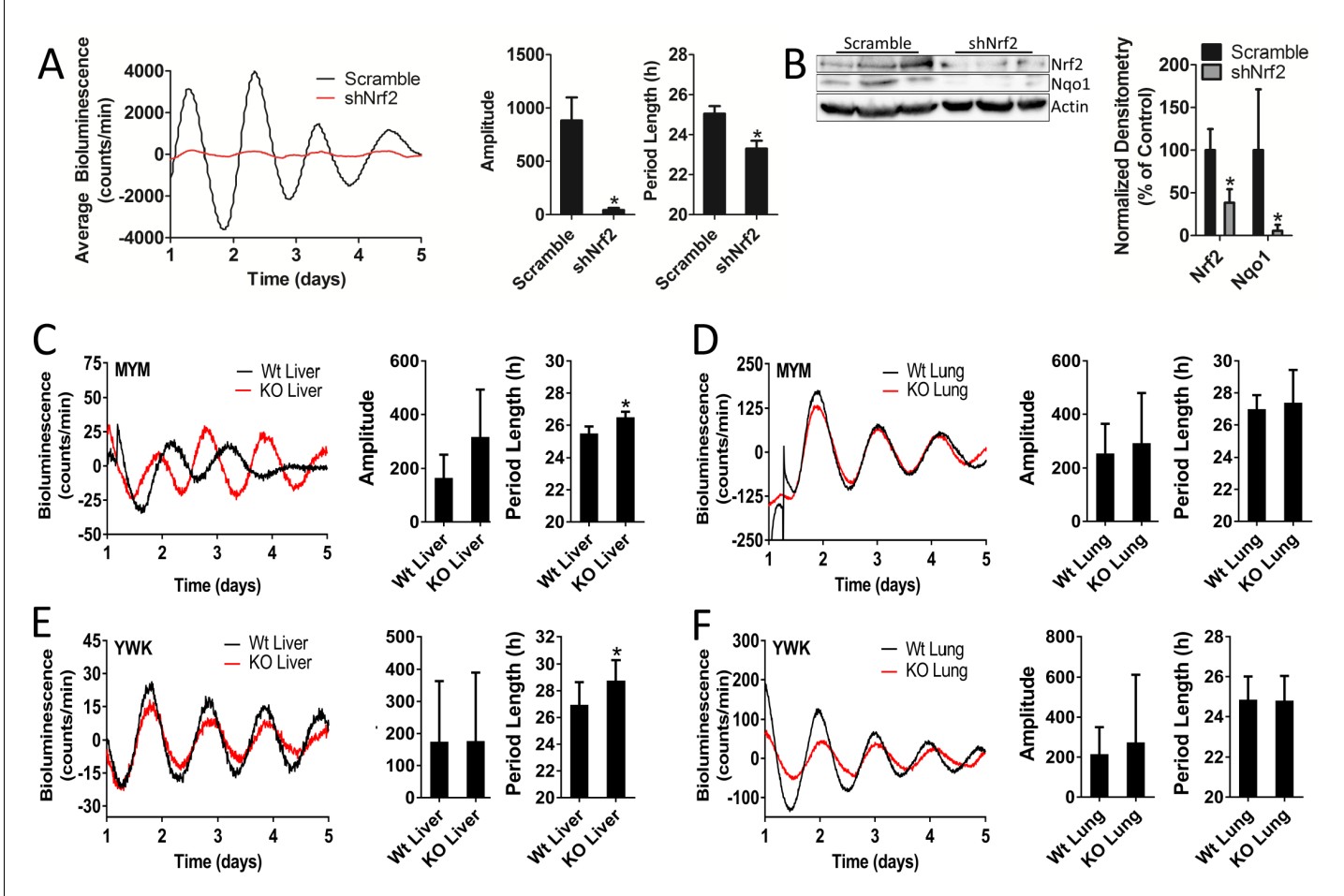

**Figure 5.** Nrf2 is required to maintain mouse hepatocyte and liver circadian pace. (**A**) Per2:Luc-driven bioluminescence from differentiated MMH-D3 hepatocytes stably expressing a scramble shRNA vector (Scramble) or shNrf2. Average luminescence recordings are shown on the left. Amplitude and period length are expressed as an average ±standard deviation (n = 4). Results were analyzed using Student's *t*-test; * indicates a p-value<0.05 relative to Scramble. (**B**) NRF2 and NQO1 protein in 50 µg of whole cell lysate from the MMH-D3 hepatocytes expressing a Scramble shRNA vector or shNrf2. β-Actin, which was unchanged by shRNA expression, was used as a loading control. Values are expressed as the normalized average densitometry ±standard deviation (n = 3) relative to Scramble, which was set to 100%. Results were analyzed using Student's *t*-test; * indicates a p-value<0.05 relative to Scramble. (**C–D**) Per2:Luc-driven bioluminescence from (**C**) liver and (**D**) lung organotypic slices from Wt and MYM strain *Nrf2-/-* (KO) mice. Representative luminescence recordings are shown on the left in each panel. Amplitude and period length are expressed as an average ±standard error mean (n = 5 animals, liver; n = 4 animals, lung; 2–4 organotypic slices/tissue). Results were analyzed using Student's *t*-test; * indicates a p-value<0.05 relative to Wt. (**E–F**) Per2:Luc-driven bioluminescence from (**E**) liver and (**F**) lung organotypic slices from Wt and YWK strain *Nrf2-/-* (KO) mice. Representative luminescence recordings are shown on the left in each panel. Amplitude and period length are expressed as an average ±standard error mean (n = 4 animals, liver; n = 8 animals, lung; 2–4 organotypic slices/tissue). Results were analyzed using Student's *t*-test; * indicates a p-value<0.05 relative to Wt.

DOI: https://doi.org/10.7554/eLife.31656.018

The following figure supplements are available for figure 5:

**Figure supplement 1.** Effect of shNrf2 on circadian rhythmicity in MMH-D3 hepatocytes.

DOI: https://doi.org/10.7554/eLife.31656.019

**Figure supplement 2.** SCN explant circadian rhythmicity in Wt and *Nrf2*-null mice.

DOI: https://doi.org/10.7554/eLife.31656.020

despite earlier literature indicating that NRF2 is a circadian output in the lung (*Pekovic-Vaughan et al., 2014*) and can be integrated into clock function in that tissue type in response to stress (*Yang et al., 2014*). Previous reports have indicated that the loss of *Nrf2* had no effect on the locomotor behavioral activity in mice (*Pekovic-Vaughan et al., 2014*). In agreement with this

observation, we observed no significant alteration in circadian rhythmicity in SCN organotypic slices from the YWK strain of *Nrf2*-null mice (*Figure 5—figure supplement 2*).

## Nrf2 regulates *Cry2* expression, indirectly repressing CLOCK/BMAL1 transcriptional activity and disrupting circadian rhythmicity in mouse hepatocytes

To better understand the mechanistic effects of NRF2 gain-of-function on the circadian clockwork, we established Wt and *Nrf2* overexpression MMH-D3 Per2:Luc hepatocyte cell lines. Similar to our observations in MEFs, *Nrf2* overexpression resulted in decreased rhythm amplitude and period length (*Figure 6A*). *Nrf2* overexpression also led to significantly elevated *Nqo1*, *Cry2*, and *Nr1d1* expression (*Figure 6B*).

Consistent with the effect of genetic gain-of-function, D3T-induced activation of NRF2 in hepatocytes caused significant increases in *Cry2*, *Nr1d1*, and *Nqo1* RNA expression (*Figure 6C*), suggesting these genes may be under NRF2 regulation. Using a position weight matrix implemented in the JAS-PAR CORE database (*Mathelier et al., 2016*) we identified putative AREs in the promoter regions of each of these genes (*Figure 3—figure supplement 1*, *Supplementary files 1–2*), supporting the hypothesis that they are indeed regulated by NRF2. Chromatin immunoprecipitation of NRF2 in D3T-treated hepatocytes demonstrated significantly elevated binding of NRF2 to novel enhancer elements in the promoters of *Cry2*, *Nr1d1*, and *Nqo1* (*Figure 6D*). In addition to these novel elements, we also observed D3T-induced enhanced binding of NRF2 to ARE elements previously characterized in *Nr1d1* (*Figure 6—figure supplement 1*) (*Yang et al., 2014*) and *Nqo1* (*Figure 6—figure supplement 1*) (*Nioi et al., 2003*).

Along with these effects on circadian gene expression, transfection of pLV7-Nrf2 into HEK293T cells was able to repress CLOCK/BMAL1-mediated transcription of Per1:Luc and synthetic E-box:Luc reporter constructs in a manner similar to what was observed in response to the transfection of *Cry1* (*Figure 6E*). These observations suggested that NRF2 may be physically interacting with the CLOCK/BMAL1 complex, a common property of core clock proteins (*Anafi et al., 2014*), leading to its repression. Attempts to characterize protein-protein interactions between NRF2 and any other clock protein using coimmunoprecipitation and two-hybrid assays, however, failed to detect any physical interaction involving NRF2 and another clock protein (data not shown). Our data indicating that NRF2 may be a regulator of *Cry2* expression, raised the hypothesis that NRF2 may repress CLOCK/BMAL1 transcription indirectly through the regulation of endogenous *Cry2* expression. In support of this hypothesis, we measured significant increases in both *Cry2* and *Nqo1* expression in HEK293T cells transfected with pLenti-Nrf2 (*Figure 6F*). To confirm that *Cry2* expression could account for a repression in CLOCK/BMAL1 transcription, we measured CLOCK/BMAL1-mediated Per1 reporter-driven bioluminescence in HEK293T cells as a function of *Cry1*, *Cry2*, or pLenti-Nrf2 expression. Expression of either *Cry1/2* or *Nrf2* resulted in significant repression of CLOCK/BMAL1 transcriptional activity (*Figure 6G*).

To evaluate whether the regulation of circadian gene expression in response to NRF2 activation manifests in changes to circadian function, we measured bioluminescence expression patterns in Per2:Luc MMH-D3 hepatocytes in response to increasing doses of D3T. Both circadian parameters of rhythm amplitude and period length decreased as a function of D3T dose escalation (*Figure 6H*), while having no measurable negative effect on cell health (*Figure 6—figure supplement 2*). Interestingly, the same D3T dose escalation that produced reductions in rhythm amplitude and period length also led to the dose-dependent induction of *Nqo1* and *Cry2* expression (*Figure 6I*).

Together, these data support the concept that NRF2 likely represses CLOCK/BMAL1-mediated transcription indirectly through the regulation of *Cry2* and *Nr1d1* expression. Given the reported E-Box-mediated circadian regulation of *Nrf2* (*Pekovic-Vaughan et al., 2014*), these results indicate that NRF2 and clock comprise an interlocking loop (*Figure 6J*) that integrates cellular redox signals into circadian timekeeping.

## Discussion

In mammals, the molecular core circadian clock mechanism is a transcription-translation feedback loop. Positive transcriptional regulation of the E-box containing genes *Per* and *Cry* by CLOCK and BMAL1 is followed by translation of PER and CRY, which heterodimerize and repress their own

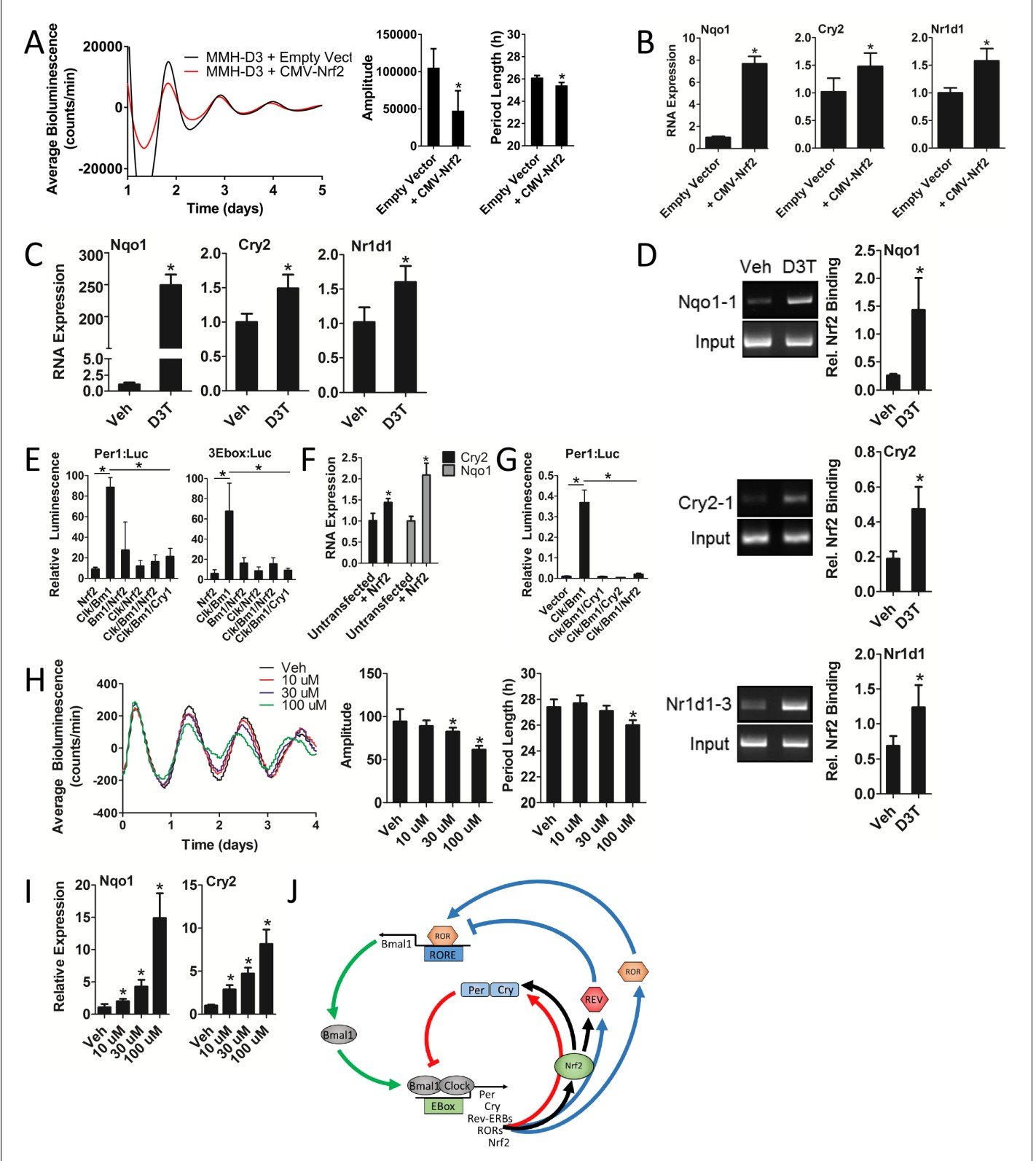

**Figure 6.** Nrf2 regulates *Cry2* expression, indirectly repressing CLOCK/BMAL1 transcriptional activity and disrupting circadian rhythmicity in mouse hepatocytes. (**A**) Per2:Luc-driven bioluminescence in differentiated MMH-D3 hepatocytes stably expressing an empty vector (Empty Vector) or a *Nrf2*-expression construct (CMV-Nrf2). Average luminescence recordings are shown on the left. Amplitude and period length are expressed as an average ±standard deviation (n = 3). Results were analyzed using Student's *t*-test; * indicates a p-value<0.05 relative to the Empty Vector. (**B**) *Nqo1*,
*Figure 6 continued on next page*

*Figure 6 continued*

*Cry2*, and *Nr1d1* gene expression in the differentiated MMH-D3 hepatocytes expressing an empty vector or CMV-Nrf2 used in 'A'. Expression values were determined by qPCR and normalized to *Gapdh*. Data is shown as the average fold-change ±standard deviation (n = 3) relative to the expression in the Empty Vector, which was set to 1. Results were analyzed using Student's *t*-test; * indicates a p-value<0.05 relative to the Empty Vector. (C) *Nqo1*, *Cry2*, and *Nr1d1* gene expression in differentiated MMH-D3 hepatocytes treated with DMSO (Veh) or 300 µM D3T for 7 hr. Expression values were determined by qPCR and normalized to *Gapdh*. Data is shown as the average fold-change ±standard deviation (n = 3) relative to the expression in Veh, which was set to 1. Results were analyzed using Student's *t*-test; * indicates a p-value<0.05 relative to Veh. (D) Chromatin immunoprecipitation-PCR using an NRF2 antibody to detect binding of NRF2 to the indicated gene promoters. Differentiated MMH-D3 hepatocytes treated with DMSO (Veh) or 300 µM D3T for 7 hr. Putative ARE sequences can be found in **Supplementary file 2** and their location in the respective promoter regions can be found in **Figure 3—figure supplement 1**. Quantitation by densitometry of NRF2 binding to the enhancers is indicated to the right of each representative picture. Data is shown as an average ±standard deviation (n = 3–5). Results were analyzed using Student's *t*-test; * indicates a p-value≤0.05 relative to Veh. (E) CLOCK/BMAL1-mediated luciferase activity generated from *Per1*- or triplicate synthetic E-box-driven (3Ebox) luciferase reporter constructs in HEK293T cells as a function of transfection of the indicated expression constructs, including pLV7-P(CMV)-Nrf2. Data are shown as the average luminescence ±standard deviation (n = 3) normalized to renilla luminescence to control for transfection efficiency. Results were analyzed using a one-way ANOVA followed by Dunnett's multiple comparisons test; * indicates a p-value<0.05 relative to the *Clock/Bmal1* (*Clk/Bm1*) positive control. (F) *Cry2* and *Nqo1* gene expression in untransfected or pLenti-Nrf2 transfected HEK293T cells. Expression values were determined by qPCR and normalized to *Gapdh*. Data is shown as the average fold-change ±standard deviation (n = 3) relative to the expression in the untransfected cells, which was set to 1. Results were analyzed using Student's *t*-test; * indicates a p-value<0.05 relative to the untransfected control. (G) CLOCK/BMAL1-mediated luciferase activity generated from a *Per1*-luciferase reporter construct in HEK293T cells as a function of transfection of the indicated expression constructs, including pLenti-Nrf2. Data are shown as the average luminescence ±standard deviation (n = 3) normalized to renilla luminescence to control for transfection efficiency. Results were analyzed using a one-way ANOVA followed by Dunnett's multiple comparisons test; * indicates a p-value<0.05 relative to the *Clock/Bmal1* (*Clk/Bm1*) positive control. (H) Per2:Luc-driven bioluminescence from differentiated MMH-D3 hepatocytes in the presence of DMSO (Veh) or D3T at the indicated concentrations. Average luminescence recordings are shown on the left. Amplitude and period length are expressed as an average ±standard deviation (n = 8). Results were analyzed using Student's *t*-test; * indicates a p-value<0.05 relative to the Veh. (I) *Nqo1* and *Cry2* gene expression in differentiated MMH-D3 hepatocytes treated with DMSO (Veh) or the indicated concentrations of D3T. Expression values were determined by qPCR and normalized to *Gapdh*. Data is shown as the average fold-change ±standard deviation (n = 3) relative to the expression in Veh, which was set to 1. Results were analyzed using Student's *t*-test; * indicates a p-value<0.05 relative to Veh. (J) Schema showing the proposed regulation of both core and stabilizing circadian clock loops in mouse hepatocytes. NRF2 appears to be tightly integrated into the clock mechanism. As a result of being an E-box-mediated output (**Pekovic-Vaughan et al., 2014**) and a transcriptional regulator of *Cry2* and *Nr1d1*, NRF2 input can both reinforce and shift the circadian phase, thus forming a redox sensitive interlocking loop.

DOI: https://doi.org/10.7554/eLife.31656.021

The following figure supplements are available for figure 6:

**Figure supplement 1.** D3T-induced NRF2 binding to AREs in the *Nr1d1* and *Nqo1* promoter regions.
DOI: https://doi.org/10.7554/eLife.31656.022
**Figure supplement 2.** D3T cytotoxicity in MMH-D3 hepatocytes.
DOI: https://doi.org/10.7554/eLife.31656.023

transcription through the inhibition of the CLOCK/BMAL1-complex. Circadian regulation of RORE- and D-box containing genes comprise additional transcription-translation feedback loops interlocked with the core clock mechanism (**Zhang and Kay, 2010**; **Takahashi, 2017**). These additional feedback loops stabilize and contribute to clock robustness and also serve as entry points for external feedback. Synchronicity between the circadian clock and the environment is maintained by entraining the clock to predictable external cues. A predominant entraining signal in peripheral tissues is diurnal metabolic cycles (**Hara et al., 2001**). As a natural consequence of oxidative metabolism, perturbations in the ratios of redox coupled cofactors and the production of oxidative signals are thought to be the metabolic inputs into the clockwork.

Redox rhythms, including oscillations in $H_2O_2$ and hyperoxidized peroxiredoxin (**Edgar et al., 2012**), exist across all domains of life and are believed to cross-talk with the molecular clock. Parallels between these redox oscillations, metabolic cycles, and circadian gene expression led to the idea that changes in the intracellular redox potential may be the link coupling metabolism to the clock (**Putker and O'Neill, 2016**). In biochemical assays, increases in the $NAD^+$/NADH ratio, to mimic an oxidized intracellular environment, resulted in reduced CLOCK/BMAL1 DNA-binding affinity (**Rutter et al., 2001**). Additionally, the levels of *zCry1* and *zPer2* were elevated in zebrafish Z3 cells in response to light-induced $H_2O_2$ production (**Hirayama et al., 2007**). A single $H_2O_2$ bolus recapitulated these effects, confirming that $H_2O_2$ was indeed a circadian input signal.

Despite these observations, the direct mechanism of redox, and by extension metabolic, input into the clock has yet to be fully elucidated. A previous study reported a NRF2-binding site in the

promoter of the clock gene *Nr1d1* and its subsequent regulation in response to $H_2O_2$ and sulforaphane (*Yang et al., 2014*), a well-characterized NRF2 activator (*Kensler et al., 2013*). As a primary sensor of intracellular oxidation, metabolic input into the clock via NRF2-mediated transcription seemed fitting. A recent report demonstrated that that genetic or chemical inhibition of the pentose phosphate pathway (PPP) altered circadian period length in a NRF2-dependent manner (*Rey et al., 2016*). Here, we substantiated these findings, showing that electrophilic or oxidative activation of KEAP1/NRF2 signaling altered clock gene expression and circadian function. Expression of E- and D-box-regulated circadian genes, including *Per3*, *Nr1d1*, *Nr1d2*, *Dbp*, and *Tef*, were elevated via a NRF2-dependent mechanism in the liver of mice treated with D3T, indicating possible NRF2 input into multiple circadian loops. These changes in the levels of RNA paralleled significant reduction of circadian rhythm amplitude in MEFs in the presence of D3T. This effect was absent in treated *Nrf2-/-* cells. Oxidative activators of KEAP1/NRF2 signaling, tBHQ and $H_2O_2$, had similar NRF2-dependent effects on rhythm amplitude. We further validated the effects on circadian rhythmicity in response to NRF2-activating chemicals using genetic activation of NRF2. Overexpression of NRF2 or deletion of its negative regulator, *Keap1*, phenocopied the effects of NRF2-activating chemicals on circadian function, in part through NRF2-mediated *Nr1d1* regulation. Together, these data support the idea that NRF2 is a critical node linking metabolism to the clock as well as a conduit for the response to changes in intracellular redox status.

As alterations in rhythm amplitude and period length are considered to be reliable indicators of clock fidelity (*Takahashi et al., 2008*), an important and heretofore unreported observation of our study is the significant alteration in circadian rhythmicity in mouse fibroblasts, hepatocytes, and liver explants lacking *Nrf2*. Along with the effects of NRF2 gain-of-function on rhythmicity, these observations indicate that the stoichiometry and/or the timing of *Nrf2* expression are important for maintaining clock function. Marked circadian disruption in the absence of NRF2 supports an endogenous role for NRF2 in timekeeping in certain tissues, as these effects were observed in liver, but not in lung or SCN explants. This lack of effect in SCN explants is consistent with a previous report demonstrating that *Nrf2-/-* mice display Wt locomotor activity patterns (*Pekovic-Vaughan et al., 2014*).

By characterizing the circadian timing of redox and NRF2 cycling, we showed that intracellular oxidation precedes NRF2 accumulation. Activation of NRF2 at a circadian time when the intracellular redox potential is decreasing leads to a NRF2-dependent enhancement of circadian amplitude, a response termed reinforcement (*Zhou et al., 2015*). Previous studies have reported that both *Keap1* expression and GSH levels are lowered prior to the production of endogenous oxidants and the induction of *Nrf2* in mouse liver (*Xu et al., 2012*). These results indicate that NRF2 expression is rapidly increasing at a time when intracellular reducing capacity is at its lowest. Decreasing intracellular redox potential promotes the accumulation of oxidants and a reduction in intracellular pH. It is therefore plausible that the simultaneous decreases in redox potential and intracellular pH lead to the oxidative modification of KEAP1 cysteine residues and the activation of NRF2. In light of these previous findings, our results showing that oxidative activation of NRF2 at this time reinforces the circadian clock to increase oscillator amplitude is consistent with tight coupling of cellular redox status, NRF2 activation, and clock function.

The NRF2-dependent reinforcement of circadian amplitude is nearly identical to that attributed to the *Arabidopsis* NPR1 (non-expressor of pathogenesis-related gene 1), a redox-sensitive modulator of clock gene expression. When salicylic acid is produced and activates NPR1 at an appropriate time, NPR1 transcriptionally regulates gene expression in the morning and evening plant circadian loops to reinforce the circadian amplitude. In addition to reinforcing circadian amplitude, NPR1 also segregates the initiation of cellular defense and immunity pathways to the morning phase of the cycle (*Zhou et al., 2015*). This provides a mechanism to maximize energy expenditure during the night phase to support plant growth and overall fitness (*Nozue et al., 2007*; *Dodd et al., 2005*). As an analog of NPR1, NRF2 may contribute to a similar temporal segregation of energy utilization pathways in animals.

In addition to amplitude reinforcement, our data indicate that physiological levels of $H_2O_2$ may cause phase-dependent circadian phase-resetting. Similar findings have been reported recently in multiple mammalian cell lines demonstrating that redox signals and the intracellular redox environment can elicit shifts in circadian phase and regulate rhythm amplitude (*Putker et al., 2018*). These findings, in combination with our observations that both *Cry2* and *Nr1d1* are transcriptionally regulated by NRF2, suggest a plausible mechanism through which endogenous oxidative signals directly

input into the circadian clockwork to link metabolism and timekeeping. Thus, while acute changes in redox balance may act as nonessential auxiliary timekeepers (*Putker et al., 2018*), proper alignment of the intracellular redox and circadian phases appears to enhance circadian function, implying the existence of a direct and reciprocal coupling between the redox state and the circadian mechanism. We believe that NRF2 is the mechanistic conduit interconnecting these two processes.

Activation of NRF2 is emerging as a key regulator of metabolism through the regulation of the PPP and by directing glutamine breakdown into glutathione biosynthetic pathways (*Mitsuishi et al., 2012*; *Hayes and Dinkova-Kostova, 2014*). NRF2-dependent regulation of the PPP suggests that NRF2 may be a contributor in the daily transition between catabolic and anabolic metabolism. NRF2 feedback into the circadian clockwork could be an effective mechanism to facilitate this transition by integrating metabolic output and intracellular redox homeostasis into rhythmicity and carbohydrate flux. Similarities between NPR1 and NRF2 in their regulation of antioxidant response and metabolic pathways, and their common integration into the circadian function of their respective organisms, indicate that the interconnectedness of these pathways is likely a product of convergent evolution. We propose that the coordinated integration of metabolism, redox homeostasis, and the circadian clock in mammals through NRF2 is a mechanism to align energy production and utilization to the environmental light-dark cycle, as has been observed in plants (*Dodd et al., 2005*).

Proper integration of redox signals into the clock requires the activation of NRF2 by levels of oxidants low enough to affect signaling without inducing damage. A recent study seeking to link metabolic redox oscillations to the clock through NRF2 relied on PPP inhibition to reduce NADPH production, increasing the $NADP^+$/NADPH ratio (*Rey et al., 2016*). However, data from this study shows that these conditions created a persistently elevated oxidizing environment, as evidenced by the hyperoxidation of PRDX, leading to the activation of a NRF2-mediated stress response and a disruption in circadian function. Similar disruptions in clock function in response to oxidative stress were observed in the lungs of newborn mice exposed to 95% $O_2$ (*Yang et al., 2014*). This hyperoxic exposure altered the timing and magnitude of E-box containing gene expression in vivo and NRF2 regulation of *Nr1d1* in cells. Interestingly, we observed unaltered circadian rhythmicity in lung explants from two independent strains of *Nrf2*-null mice, suggesting that NRF2 signaling is not required for normal circadian function in the lung. Nonetheless, the reports showing that NRF2 alters circadian *Nr1d1* expression and the disruption of cellular clock function by persistent oxidative activation of NRF2 (*Yang et al., 2014*) indicates that ROS-activated NRF2 signals into the clock mechanism. Thus, it appears that NRF2 and the stress-response it regulates are at the top of a molecular hierarchy whereby the resolution of oxidative stress likely takes precedence over the preservation of timekeeping. However, because the circadian clock is reset daily by environmental and systemic zeitgebers, stress-induced disruption of circadian function is likely to be transient and unlikely to manifest in deleterious effects on cellular or tissue health.

In hepatocytes, where NRF2 and the clock appear to be tightly coupled, we found that activated NRF2-bound specific enhancer regions of the core clock repressor gene *Cry2*, increased *Cry2* expression, and repressed CLOCK/BMAL1-regulated E-box transcription. Moreover, activation of NRF2 in these cells also resulted in dose-dependent decreases in rhythm amplitude and period length, with associated dose-dependent increases in *Cry2*. Circadian amplitude and period length are largely dependent on CRY repression of CLOCK/BMAL1-mediated E-box gene transcription, placing the regulation of *Cry* at the heart of the clock mechanism (*Sato et al., 2006*; *Kume et al., 1999*; *Reppert and Weaver, 2002*; *Griffin et al., 1999*). Despite its role in the clockwork, little is known about the transcriptional regulation of *Cry2*. Our characterization of a NRF2-binding site in the *Cry2* promoter, in addition to its induction, supports the idea that NRF2-dependent regulation of *Cry2* is a mechanism to relay timing information from the redox oscillator into the molecular clock. Interestingly, circadian regulation of metabolism also occurs via this same mechanism. Beyond being NRF2-dependent E-box repressors, both CRY2 and NR1D1 are also important regulators of nuclear hormone receptors, governing lipid, glucose, and xenobiotic metabolism (*Zhang et al., 2015*; *Kriebs et al., 2017*). Additional NRF2-dependent regulation of the production of reducing equivalents through the PPP (*Mitsuishi et al., 2012*) and heme levels through heme oxygenase one expression (*Ishii et al., 2000*), which serve as cofactors for CRY2 and NR1D1, respectively, further integrate metabolism and the clock mechanism and highlight the interconnectedness of these oscillators.

While the principle that metabolism regulates the clock through oscillations in the redox balance is generally accepted (*Asher and Schibler, 2011*), the molecular mechanisms through which the

redox state is integrated into the clockwork is just emerging. As an E-box-regulated circadian output (*Pekovic-Vaughan et al., 2014*), the data presented here suggest that NRF2 likely represses its own transcription through the regulation of *Cry2* and subsequent CRY2-mediated repression of CLOCK/BMAL1, thus forming a redox-responsive transcription-translation feedback loop interlocked with the core molecular circadian mechanism. From these observations, NRF2 appears to be a key mechanistic link between circadian oscillations in redox balance and clock gene expression rhythmns. Reciprocal integration of redox potential and the clock through NRF2-dependent regulation reinforces the efficiency of both the circadian and metabolic oscillators, contributing to enhanced organismal fitness.

# Materials and methods

**Key resources table**

| Reagent type (species) or resource | Designation | Source or reference | Identifiers | Additional information |
|---|---|---|---|---|
| Gene (*Mus musculus*) | Nfe2l2 | NA | MGI:108420 | |
| Strain, strain background (*M. musculus*) | YWK | PMID: 8943040 | Nrf2-/-; YWK Strain | |
| Strain, strain background (*M. musculus*) | MYM | PMID: 9240432 | Nrf2-/-; MYM Strain | |
| Strain, strain background (*M. musculus*) | PER2::LUC | PMID: 14963227 | PER2::LUC | |
| Genetic reagent (*M. musculus*) | Per2:Luc | PMID: 23052244; 24699442; 18454201; 17482552 | Per2:Luc | |
| Cell line (*M. musculus*) | Wt MEF Per2:Luc | this paper | Wt MEF Per2:Luc | Immortalized Wt MEF cell line stably expressing a Per2:Luc reporter. Created by Thomas R. Sutter's Lab at the University of Memphis. |
| Cell line (*M. musculus*) | Nrf2-/- MEF Per2:Luc | this paper | Nrf2-/- MEF Per2:Luc | Immortalized Nrf2-/- MEF cell line stably expressing a Per2:Luc reporter. Created by Thomas R. Sutter's Lab at the University of Memphis. |
| Cell line (*M. musculus*) | Keap1-/- MEF Per2:Luc | this paper | Keap1-/- MEF Per2:Luc | Immortalized Keap1-/- MEF cell line stably expressing a Per2:Luc reporter. Created by Thomas R. Sutter's Lab at the University of Memphis. |
| Cell line (*M. musculus*) | MMH-D3 Per2:Luc | PMID: 24699442 | MMH-D3 Per2:Luc | |
| Antibody | anti-Nrf2 (rabbit polyclonal) | Santa Cruz | Santa Cruz SC-722 | (1:500) |

*Continued on next page*

*Continued*

| Reagent type (species) or resource | Designation | Source or reference | Identifiers | Additional information |
|---|---|---|---|---|
| Antibody | anti-Nqo1 (rabbit monoclonal) | Abcam | Abcam AB80588 | (1:10,000) |
| Antibody | anti-Nr1d1 (rabbit monoclonal) | Cell Signaling | Cell Signaling 13418 | (1:500) |
| Antibody | anti-Prdx-SO2/3 (rabbit polyclonal) | Thermo Fisher | Thermo LF-PA0004 | (1:1000) |
| Antibody | anti-Per2 (rabbit polyclonal) | ProteinTech | ProteinTech 20359 | (1:200) |
| Antibody | anti-Nrf2 (rabbit polyclonal) | Diagenode | Diagenode C15410242 | (2 ug) |
| Recombinant DNA reagent | shNrf2 (shRNA) | Sigma | Sigma NM_010902:TRCN0000012128 | |
| Recombinant DNA reagent | shNrf2 (shRNA) | this paper | shNrf2-1 | shNrf2 lentiviral vector was developed by Andrew Liu at the University of Memphis. Target sequence is described in *Supplementary file 4*. |
| Recombinant DNA reagent | shNrf2-2 (shRNA) | this paper | shNrf2-2 | shNrf2 lentiviral vector was developed by Andrew Liu at the University of Memphis. Target sequence is described in *Supplementary file 4*. |
| Recombinant DNA reagent | shNrf2-3 (shRNA) | this paper | shNrf2-3 | shNrf2 lentiviral vector was developed by Andrew Liu at the University of Memphis. Target sequence is described in *Supplementary file 4*. |
| Recombinant DNA reagent | CMV-Nrf2 (Expression vector) | this paper | CMV-Nrf2 | CMV-driven Nrf2 expression construct was created in the pLV7 lentiviral vector backbone using Gateway recombination. Primers to amplify Nrf2 cDNA and shuttle the ORF into pLV7 are described in *Supplementary file 3*. |
| Recombinant DNA reagent | pLENTI-Nrf2 (Expression vector) | Origene | Origene MR226717L1 | |
| Recombinant DNA reagent | pGL3-P(Per1)-dLuc | PMID: 22692217 | Per1:Luc | |
| Recombinant DNA reagent | pGL3-3xEbox-P(SV40)-dLuc | PMID: 22692217 | 3xEbox:Luc | |
| Commercial assay or kit | Lenti-X p24 Rapid Titer Kit | Clontech | Clontech 632200 | |
| Chemical compound, drug | Hydrogen peroxide | Sigma | Sigma 216763 | |
| Chemical compound, drug | D3T | LKT Laboratories | LKT Laboratories D0010 | |
| Software, algorithm | LumiCycle Analysis | Actimetrics | Version 2.31 | |

## Chemicals

CDDO-Im was obtained from Dr. Michael B. Sporn at Dartmouth Medical School. D3T and $H_2O_2$ were purchased from LKT Laboratories (St. Paul, MN) and Sigma (St. Louis, MO), respectively. The integrity and concentration of hydrogen peroxide was verified prior to each experiment by spectroscopic detection at 240 nm, using an extinction coefficient of 43.6 $M^{-1}$ $cm^{-1}$ (*Noble and Gibson, 1970*). Vehicle treatments for all experiments contained either water or DMSO as indicated in the figure legends. The concentration of DMSO never exceeded 0.05%.

## Animals and treatment

All experiments were approved by the University of Memphis Institutional Animal Care and Use Committee and carried out in accordance with the Guide for the Care and Use of Laboratory Animals as adopted and promulgated by the U.S. National Institutes of Health. Male mice ranging from 3 to 6 months of age were used for all experiments. Animals were housed in a temperature and humidity controlled room with a 12:12 hr light:dark cycle. Food and water were provided *ad libitum*. *Nrf2-/-* mice were developed by Dr. Yuet Wai Kan (YWK strain) (*Chan et al., 1996*) and Dr. Masayuki Yamamoto (MYM strain) (*Itoh et al., 1997*) and obtained from Jackson Laboratories (Bar Harbor, ME) or from Dr. Thomas Kensler at the University of Pittsburgh, respectively. PER2::LUC transgenic mice, originally generated by Dr. Joe Takahashi (*Yoo et al., 2004*), were obtained from Jackson Laboratories. Wt and *Nrf2-/-* PER2::LUC mouse strains were created by crossing Wt C57BL/6J and *Nrf2-/-* C57BL/6J mice with PER2:LUC transgenic mice. Offspring heterozygous for both alleles were crossed and homozygous littermates were used for each experiment. All genotypes were confirmed using PCR amplification of tail genomic DNA (*Supplementary file 3*). For gene expression studies, animals were treated by gavage with vehicle [10% Cremophor EL (Sigma), 10% dimethyl sulfoxide, 80% phosphate-buffered saline] or 300 µmol/kg bw D3T every other day for 5 days. Mice were euthanized 24 hr following the final treatment. Liver tissue was excised and snap frozen.

## Per2:Luc MEF bioluminescence assays

Stable Per2:Luc reporter MEF cell lines were created by infecting MEFs of each genotype with lentiviral particles (*Ramanathan et al., 2012*) carrying a vector expressing a Per2-driven luciferase reporter construct, as described (*Liu et al., 2008*; *Liu et al., 2007b*). Reporter cell lines were selected by the addition of 0.01 mg/mL blasticidin added to the growth medium. Genotypes were confirmed by PCR (*Supplementary file 3*). All cultures were checked and confirmed to be free from mycoplasma using the MycoSensor PCR Kit (Agilent, Santa Clara, CA). All MEF cell lines were maintained in DMEM high glucose (Fisher Scientific, Hampton, NH, 10–013-CV) medium supplemented with 10% fetal bovine serum and 1x penicillin/streptomycin (complete growth medium). 2 µg/mL puromycin was added to the growth medium for all MEF cell lines transduced with shRNA or *Nrf2* overexpression lentiviral vectors. Cells were cultured to confluence on collagen-coated 35 mm cell culture plates. At confluence, the growth medium was removed, the cells were washed once with PBS, and freshly made Recording Medium (*Yoo et al., 2004*; *Yamazaki and Takahashi, 2005*) containing 2% FBS, 2% B27 (Thermo Fisher, Waltham, MA), and 1 mM luciferin was added to the cultures. Plates were sealed with a vacuum grease coated cover slip and loaded into a Lumicycle 32 (Actimetrics, Wilmette, IL) housed in a non-humidified 37°C $CO_2$-buffered incubator for real-time bioluminescence recording. The addition of Recording Medium was sufficient to synchronize the cell population producing coherent luminescence signals. For assays evaluating the effects of D3T, tBHQ, $H_2O_2$, and CDDO-Im, each treatment was made in Recording Medium and added to the plates at the time they were loaded into the Lumicycle. For treatment-induced amplitude reinforcement assays, plates were removed from the Lumicycle at the indicated time, $H_2O_2$ or D3T was added to the existing medium at the times indicated, the plates were resealed with cover slips, and the dishes were loaded back into the Lumicycle. We did not observe measurable deviations in bioluminescence independent of treatment following this procedure.

## Per2:Luc MMH-D3 hepatocyte bioluminescence assays

MMH-D3 Per2:Luc reporter cell lines were provided by Dr. Andrew Liu at the University of Memphis and confirmed to be free from mycoplasma using the MycoSensor PCR Kit (Agilent). MMH-D3 hepatocytes were cultured in RPMI1640 medium (Fisher Scientific, 11875) supplemented with 10% fetal

bovine serum, 1x penicillin/streptomycin, 10 µg/ml insulin, 55 ng/ml epidermal growth factor, and 16 ng/ml insulin-like growth factor-II to confluence in collagen-coated 35 mm cell culture plates. At confluence, the hepatocytes were differentiated by adding 2% (v/v) DMSO to the growth medium. Differentiation medium was refreshed every other day for 5 days. Following differentiation, cells were washed with PBS and serum-free growth medium containing 200 nM dexamethasone was added for synchronization. Cells were incubated in dexamethasone-containing medium for 2 hr in a 37°C humidified and $CO_2$-buffered cell culture incubator. Following synchronization, cells were washed once with PBS, and freshly made Recording Medium containing 2% FBS, 2% B27, and 1 mM luciferin was added. Plates were sealed with vacuum-coated coverslips and loaded into a Lumicycle 32 for real-time bioluminescence recording. For high-throughput luminescence studies, MMH-D3 hepatocytes were cultured in collagen-coated 96-well plates, differentiated, and synchronized as described above. Bioluminescence was recorded using a Synergy 2 SL microplate reader (Bio Tek, Winooski, VT), as previously described (*Ramanathan et al., 2012*).

## Per2:Luc organotypic slice bioluminescence assays

Liver, lung, and SCN organotypic slices were prepared as previously described (*Liu et al., 2007b*). Briefly, Wt and *Nrf2-/-* mice harboring at least one copy of the PER2::LUC transgene were euthanized and dissected to remove the liver, lung, and SCN. The liver and lung were washed in ice cold PBS and sliced by hand on ice with a razor blade in approximately 2 mm x 2 mm square pieces. Liver and lung slices were washed again with Recording Medium and placed directly into 35 mm culture dishes. The SCN sections were washed with ice cold HBSS and sliced using a vibrating microtome. SCN slices were then washed with Recording Medium and cultured on Millipore membrane inserts (Merck Millipore, Billerica, MA, PICM0RG50), in 35 mm cell culture plates. Recording Medium (900 µL) was added to dishes containing each tissue slice. Dishes were sealed with a vacuum grease-coated cover slip and loaded into a Lumicycle for bioluminescence recordings.

## Bioluminescence recording analysis

Bioluminescence data were analyzed using the LumiCycle Analysis software (Actimetrics, Version 2.31). Baseline subtraction was performed to detrend traces using a 24 hr moving average. Baseline subtracted bioluminescence data from cells and tissues were least-square fit to a damped sine wave from which the amplitude and period were determined as described previously (*Ramanathan et al., 2014*). Amplitude and period were determined from data collected between days ~ 1.5 and 5, with the exception of data shown in *Figure 4B–F* in which amplitude and period were determined following the addition of treatment until the end of the recording. All fittings of the data had a goodness of fit >85% for cells and >80% for tissues. For cell and tissue slice cultures, any culture that did not produce sustained rhythmic bioluminescence as defined by the inability to fit a sine wave with >80% goodness of fit to the baseline subtracted data was deemed to be an outlier and removed from the analysis. Due to transient spikes in luminescence immediately following a change of media, the first 24 hr of recording data was excluded from period length and amplitude determination.

## Stable cell line generation

shNrf2 Per2:Luc MEF cell lines were generated using infectious lentiviral particles. Infectious lentiviral particles containing shRNA constructs targeting Nrf2 or an empty vector backbone were purchased from Sigma (St. Louis, MO, SHC001V, SHCLNV-NM_010902:TRCN0000012128, *Supplementary file 4*). MEFs were infected at an MOI = 3, following the manufacturer's protocol. Briefly, viral particles plus polybrene (8 µg/ml) were added to the complete cell culture growth medium 1 day after seeding the MEFs. Cells were incubated in viral containing medium overnight in a 37 °C cell culture incubator. The next day, the cells were washed with PBS and the medium was changed to complete growth medium without viral particles. Two days post-infection, cells were selected with complete growth medium containing 2 µg/ml puromycin. Puromycin-containing medium was replaced every other day for 4 days. By the fifth day the non-infected control cell population was no longer viable. Generation of shNrf2 knockdown MMH-D3 hepatocytes was performed following a similar protocol using viral particles containing either a nonspecific sequence (Scramble) or *Nrf2* shRNA target sequences (*Supplementary file 4*).

*Nrf2* overexpression or genetic rescue cell lines were generated following published protocols (*Ramanathan et al., 2014*; *Ramanathan et al., 2012*). An *Nrf2* lentiviral expression vector was created by amplifying *Nrf2* cDNA using primers to install a 5'-CACC leader sequence on the amplicon (*Supplementary file 3*). The *Nrf2* PCR product was then cloned into pENTR/D-TOPO following the manufacturer's instructions (Invitrogen, Carlsbad, CA, K2400). The resulting pENTR-Nrf2 vector and a pENTR-CMV promoter vector were moved into a pLV7-Puro destination vector (*Liu et al., 2008*) through a multisite Gateway recombination reaction to generate the pLV7-CMV-Nrf2 lentiviral expression vector. Viral particles were generated following standard protocols in 293 T cells, as described previously (*Tiscornia et al., 2006*) with the exception that the transfection of 293T was performed using Lipofectamine 2000 (Invitrogen) in place of the calcium phosphate method described. Viral titers were determined by p24 ELISA assay using the Lenti-X p24 Rapid Titer Kit (Clontech, Mountain View, CA, 632200). Viral titers of $>10^7$ were normally achieved. Cells were infected at an MOI = 3–5 and selected with 2 μg/ml puromycin.

## RNA isolation and qPCR

To isolate RNA from mouse liver approximately 500 mg of tissue was homogenized in 1 mL RNA STAT-60 (Tel-Test, Friendswood, TX). RNA was isolated from cells by adding RNA STAT-60 directly to the cell culture plate. In both instances, RNA was extracted from the STAT-60 solution by the addition of chloroform:isoamyl alcohol (24:1). Extracted RNA was dissolved in water, reprecipitated using sodium acetate and isopropanol, washed with 75% ethanol, and quantified using a Nanodrop. For qPCR measurement of RNA transcripts, 1 μg of RNA was reverse transcribed to cDNA. 6 ng of cDNA was mixed with forward and reverse primers for the intended gene target (*Supplementary file 5*) and ABsolute Blue SYBR Green qPCR master mix (Thermo Fisher Scientific). Gene expression measurements were normalized to the expression of the endogenous reference gene, *Gapdh*, which not affected by any of the treatments or genetic manipulations.

## Western blotting

Cells grown to 100% confluence were washed with ice cold PBS and harvested by scraping in cold PBS. Cell pellets were lysed in RIPA buffer [25 mM Tris pH 7.4, 150 mM NaCl, 0.1% SDS, 0.5% sodium deoxycholate, 1% Triton X-100, and freshly added Protease Inhibitor Cocktail (Sigma), 1 mM phenylmethylsulfonyl fluoride, and 1 mM $Na_3VO_4$]. Lysates were cleared by centrifugation at 13,000 g, 10 min, 4°C. Total protein was quantified by Micro BCA Assay (Thermo Fisher). 50 μg of total protein was resolved on 10% SDS polyacrylamide gels. For all western assays, protein was transferred to PVDF membranes. Antibodies used are as follows: rabbit anti-NRF2 (Santa Cruz, SC-722; Proteintech, 16396); rabbit anti-NQO1 (Abcam, AB80588); rabbit anti-NR1D1 (Cell Signaling, 13418). Following incubation with primary antibody, blots were incubated with the appropriate horseradish peroxidase-conjugated secondary antibodies for 1 hr at room-temperature. Clarity (Bio-Rad, Hercules, CA) substrate was used for chemiluminescent detection using a ChemiDoc (Bio-Rad).

## Protein timecourse assays

Wt and *Nrf2-/-* Per2:Luc MEFs were cultured to confluence in complete growth medium. Cells were washed once with PBS and changed into serum-free growth medium containing 200 nM dexamethasone and incubated for 2 hr in a standard cell culture incubator for synchronization. Following synchronization, cells were washed once with PBS and complete growth medium was added (t = 0). Cells were harvested at the indicated times post-synchronization as described above. MMH-D3 hepatocytes were cultured, differentiated, and synchronized following the protocol described for the bioluminescence assays. Following synchronization, Recording Medium was added to begin the timecourse. Samples were harvested every 4 hr for 52 hr. For all timecourse assays, 50 μg of total protein extract was resolved on 10% polyacrylamide gels and transferred to PVDF membranes. Membranes were incubated with rabbit anti-PRDX-SO2/3 (Thermo LF-PA0004); rabbit anti-PER2 (Proteintech, 20359); rabbit anti-NRF2 (Santa Cruz, SC-722); and rabbit anti-NR1D1 (Cell Signaling, 13418).

## Chromatin immunoprecipitation (ChIP)-PCR assays

ChIP-PCR was performed as described previously (*Sutter et al., 2009*). Lysates from 2 × 150 mm dishes were sonicated in 1 ml of lysis buffer (1% SDS, 10 mM EDTA, 50 mM Tris-HCl, ph 8) using the Covaris

S220 (35 min, 147 Watts, 200 cycle/burst, duty factor of 5) and tubes containing an AFA fiber (Covaris, Woburn, MA). NRF2 was immunoprecipitated from the lysate using rabbit anti-NRF2 antibodies (Diagenode, Denville, NJ, C15410242-100; Santa Cruz, SC-722). Genomic DNA sequence 3.5 kb upstream and 1 kb downstream of the *Cry2*, *Nr1d1*, and *Nqo1* transcriptional start sites were analyzed for putative ARE elements using a position weight matrix implemented in JASPAR (*Mathelier et al., 2016*). ARE sequences characterized by JASPAR in the mouse genome reference GRCm38.p4 are located in *Supplementary file 2*. Primers were designed (*Supplementary file 1*) to flank the putative ARE sequences with a relative profile score of 79% or better.

## HEK293T transfection, gene expression, and in vitro luciferase transcription repression assay

HEK293T cells were grown in DMEM high-glucose (Fisher Scientific) medium supplemented with 10% fetal bovine serum and 1x penicillin/streptomycin. HEK293T cells seeded in six well plates were transfected with 25 ng/well pLenti-Nrf2 (OriGene, Rockville, MD, MR226717L1) using Lipofectamine 2000 following the manufacturer's protocol and incubated overnight at 37°C in a $CO_2$-buffered incubator. Following overnight incubation, the transfection medium was replaced with fresh growth medium. 48 hr post-transfection untransfected and transfected cells were harvested in RNA STAT-60. RNA was extracted, reverse transcribed, and used in a qPCR reaction to amplify either *Cry2* or *Nqo1* as described above. For in vitro transcription repression assays, approximately 15,000 cells/well were seeded in a 384-well plate. Cells in every well were transfected with 12.5 ng of either pGL3-P(Per1)-dLuc or pGL3-3xEbox-P(SV40)-dLuc reporter constructs, as indicated, and 2.5 ng of a phRL-SV40 plasmid expressing Renilla luciferase using Lipofectamine 2000 following the manufacturer's protocol. 25 ng of each of the following expression plasmids were cotransfected as indicated: pLV7-P(CMV)-Nrf2 (*Figure 6E*), pLenti-Nrf2 (*Figure 6G*), pLV7-P(CMV)-Bmal1, pLV7-P(CMV)-Clock, pCMV10-P(CMV)-Cry1, or pCMV10-P(CMV)-Cry2. Empty pCMV10 vector was used to normalize the total amount of DNA/well to 90 ng. 48 hr post-transfection, cells were harvested and assayed with the Dual-Glo Luciferase Assay System (Promega, Madison, WI). Firefly luminescence was normalized to Renilla luminescence as an internal control for transfection efficiency.

## Acknowledgements

We wish to thank Dr. Yang Shen for helpful conversations, technical assistance, and insight he provided on this work, as well as Dr. Zibiao Guo for his technical expertise and assistance in chromatin immunoprecipitation. We also thank Dr. David Ferster (Actimetrics) for helpful conversations and information regarding the software analysis package included with the LumiCycle.

## Additional information

### Funding

| Funder | Grant reference number | Author |
| --- | --- | --- |
| National Institutes of Health | R01 ES017014 | Thomas R Sutter |
| University of Memphis W. Harry Feinstone Center for Genomic Research | FCGR Graduate Assistantship | Ryan S Wible |
| National Institutes of Health | R01 NS054794 | Andrew C Liu |
| National Institutes of Health | R01 CA197222 | Thomas W Kensler |

The funders had no role in study design, data collection and interpretation, or the decision to submit the work for publication.

### Author contributions

Ryan S Wible, Carrie Hayes Sutter, Thomas R Sutter, Conceptualization, Data curation, Formal analysis, Investigation, Methodology, Writing—original draft, Writing—review and editing; Chidambaram Ramanathan, Andrew C Liu, Conceptualization, Data curation, Formal analysis, Investigation,

Methodology, Writing—original draft; Kristin M Olesen, Formal analysis, Investigation, Methodology, Writing—original draft; Thomas W Kensler, Conceptualization, Formal analysis, Methodology, Writing—original draft

## Author ORCIDs

Ryan S Wible ⓘ https://orcid.org/0000-0003-3326-8612
Thomas R Sutter ⓘ http://orcid.org/0000-0001-8294-7975

## Ethics

Animal experimentation: This study was performed in strict accordance with the recommendations in the Guide for the Care and Use of Laboratory Animals of the National Institutes of Health. All of the animals were handled according to approved institutional animal care and use committee (IACUC) protocols of the University of Memphis, with Animal Welfare Assurance Number A-3919-01

## Decision letter and Author response

Decision letter https://doi.org/10.7554/eLife.31656.035
Author response https://doi.org/10.7554/eLife.31656.036

## Additional files

### Supplementary files

• Supplementary file 1. Primer Pairs Used in Chip-PCR.

DOI: https://doi.org/10.7554/eLife.31656.024

• Supplementary file 2. Putative ARE enhancer elements identified by JASPAR.

DOI: https://doi.org/10.7554/eLife.31656.025

• Supplementary file 3. Genotyping and Nrf2 cloning primer pairs.

DOI: https://doi.org/10.7554/eLife.31656.026

• Supplementary file 4. shRNA target sequences.

DOI: https://doi.org/10.7554/eLife.31656.027

• Supplementary file 5. qPCR primer pairs.

DOI: https://doi.org/10.7554/eLife.31656.028

• Transparent reporting form

DOI: https://doi.org/10.7554/eLife.31656.029

### Major datasets

The following previously published dataset was used:

| Author(s) | Year | Dataset title | Dataset URL | Database, license, and accessibility information |
|---|---|---|---|---|
| Thomas R Sutter | 2017 | Pharmacogenomic comparison between D3T and CDDO-Im in mouse liver tissue | https://www.ncbi.nlm.nih.gov/geo/query/acc.cgi?acc=GSE99199 | Publicly available at the NCBI Gene Expression Omnibus (accession no. GSE99199) |

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
