## [Decision Letter]

Thank you for submitting your article "NRF2 regulates core and stabilizing circadian clock loops, coupling redox and timekeeping in mouse hepatocytes" for consideration by *eLife*. Your article has been reviewed by three peer reviewers, and the evaluation has been overseen by a Reviewing Editor and Philip Cole as the Senior Editor. The following individuals involved in review of your submission have agreed to reveal their identity: John S O'Neill (Reviewer #1); Annie Curtis (Reviewer #3).

The reviewers have discussed the reviews with one another and the Reviewing Editor has drafted this decision to help you prepare a revised submission.

Summary:

This is a well-written manuscript. The experiments are well-conceived and timely: seeking to extend current understanding a putative functional interaction between Nrf2-mediated redox relay signaling and the regulation of circadian gene expression cycles.

The major advances are that NRF2 is identified as a bona fide transcriptional regulator or Nr1d1 and Cry2, across several cell/tissue types. Genetic and pharmacological manipulations of NRF2 function have subtle effects upon timekeeping function that vary between cell types, whereas effects upon the apparent amplitude of circadian gene expression cycles are more consistent.

Most strikingly, Nrf2 appears to be essential for phase resetting in response to physiological concentrations of H_2_O_2_ in mouse embryonic fibroblasts, but surprisingly the authors have chosen to present this striking result in a supplementary figure and make little mention of it in the text.

Whilst there are many positives about this work, and there are a number of technical issues that need to be addressed. Although numerous, we believe the vast majority can be addressed readily by re-analysis of existing data and/or changes to the manuscript text.

Essential revisions:

Bioluminescence recording analysis:

The manuscript methods state: "Baseline subtracted bioluminescence signals from cells and tissues were fit to a sine wave (polynomial = 3)". The problem with detrending using any polynomial function is that, without careful examination of each interpolated baseline fit, there exists the potential to introduce non-linear trends into the detrended data, which were not present in the original recording. This has clearly occurred with some of the bioluminescence data presented in the manuscript i.e. sinusoidal data from which the baseline has been successfully subtracted should, of course, oscillate around y=0.

See Figure 1 Wt Veh for an example, the curve spends the last 2 days of the recording above 0, whereas in Figure 1 the *Nrf2-/-* Veh curve spends the last 3 days of recording below y=0. There are many more instances, with Figure 1—figure supplement 1 WT CDDO being another clear example of inappropriate detrending.

Due to the obvious artifacts introduced by this method of baseline subtraction I am not confident that sine wave fits to the detrended data can be relied upon for the estimation of any circadian parameters such as amplitude, period etc. I strongly recommend that the authors try to confirm the results of their analysis using an alternative means of baseline subtraction, such as moving average or else higher order polynomials that more accurately model the baseline changes in their data.

So that readers can judge the fidelity of baseline subtraction for themselves, perhaps the authors should present all (not just representative) raw bioluminescence traces as supplementary data along with the baseline they have subtracted in each case.

Changes in amplitude/luciferin concentration:

The apparent amplitude, phase and period of circadian rhythms reported bioluminescently is quite sensitive to the kinetics of luciferin transport over the plasma membrane, unless intracellular luciferin concentration is in saturating excess (Patrick et al., 2014; Feeney et al., 2016). Please could the authors report explicitly the concentration of extracellular luciferin employed during their assays and confirm that this is in excess i.e. addition of more luciferin extracellularly will elicit no further increase in bioluminescence if luciferin is not rate-limiting for enzyme activity.

Biological effect size:

Possibly related to the above points, I note that the difference in circadian period between wild type cells between experiments was as much as 2 hours (compare Figure 2 Wt with Figure 2 Wt or any Wt controls in Figure 4, for example). Given this large σ, a power calculation indicates that n=3 should be underpowered to detect small differences in circadian period as reported in Figure 4 for example. Could the authors please report in the methods section whether experiments that showed small (<1 h), but significant, differences in period were repeated on a separate occasion with the same result.

I also noticed that large differences in amplitude are also observed between experiments for Wt controls (see Figure 2 for example). Given that the wild type controls in 2A vs. 2B are effectively experimental repeats of each other, I do not understand why the amplitude differs by an order of magnitude. Could the authors please comment?

[Editors' note: further revisions were requested prior to acceptance, as described below.]

Thank you for resubmitting your work entitled "NRF2 regulates core and stabilizing circadian clock loops, coupling redox and timekeeping in Mus musculus" for further consideration at *eLife*. Your revised article has been favorably evaluated by Philip Cole (Senior editor), a Reviewing editor, and two reviewers.

This revised manuscript is an enormous improvement on the original submission and should be accepted for publication pending addressing two remaining issues. There are two places where the description of experimental findings might be seen as slightly misleading:

1) Taken in isolation, the abstract suggests that the authors have identified a general NRF2-dependent mechanism by which redox balance is communicated to modulate clock gene expression rhythms. This is not quite true, since there was no effect of NRF2 deletion on lung or SCN explants. It would be less misleading to add "in some cell types" to the end of the third sentence of the abstract. This in no way diminishes these important findings and is a more accurate summation of their findings.

2) The penultimate sentence of the discussion is heavily metaphor laden and could readily be misconstrued by readers outside the circadian field. Both of the terms "transcriptional clock" and "metabolic redox oscillator" suggest self-sufficiency – which is unsubstantiated. Again, it would be more accurate and lucid to amend to something like: "…NRF2 appears to be a key mechanistic link between circadian oscillations in redox balance and clock gene expression rhythms."

---

## [Author Response]

Summary:This is a well-written manuscript. The experiments are well-conceived and timely: seeking to extend current understanding a putative functional interaction between Nrf2-mediated redox relay signaling and the regulation of circadian gene expression cycles.The major advances are that NRF2 is identified as a bona fide transcriptional regulator or Nr1d1 and Cry2, across several cell/tissue types. Genetic and pharmacological manipulations of NRF2 function have subtle effects upon timekeeping function that vary between cell types, whereas effects upon the apparent amplitude of circadian gene expression cycles are more consistent.Most strikingly, Nrf2 appears to be essential for phase resetting in response to physiological concentrations of H_2_O_2_ in mouse embryonic fibroblasts, but surprisingly the authors have chosen to present this striking result in a supplementary figure and make little mention of it in the text.

We are pleased that the reviewers found our experiments well designed and our results both novel and interesting. Of interest, our lack of emphasis on the phase resetting results described above and our issues with data analysis are not simply coincidental. Because of poor baseline correction and the lower signal of MEFs, we were less confident in the reproducibility of these results. Once we became proficient in data analysis, our confidence in many of our unpublished replication experiments increased dramatically. Thus, we have revised the manuscript as suggested above in relation to phase resetting.

Essential revisions:Bioluminescence recording analysis:The manuscript methods state: "Baseline subtracted bioluminescence signals from cells and tissues were fit to a sine wave (polynomial = 3)". The problem with detrending using any polynomial function is that, without careful examination of each interpolated baseline fit, there exists the potential to introduce non-linear trends into the detrended data, which were not present in the original recording. This has clearly occurred with some of the bioluminescence data presented in the manuscript i.e. sinusoidal data from which the baseline has been successfully subtracted should, of course, oscillate around y=0.See Figure 1 Wt Veh for an example, the curve spends the last 2 days of the recording above 0, whereas in Figure 1 the Nrf2-/- Veh curve spends the last 3 days of recording below y=0. There are many more instances, with Figure 1—figure supplement 1 WT CDDO being another clear example of inappropriate detrending.Due to the obvious artifacts introduced by this method of baseline subtraction I am not confident that sine wave fits to the detrended data can be relied upon for the estimation of any circadian parameters such as amplitude, period etc. I strongly recommend that the authors try to confirm the results of their analysis using an alternative means of baseline subtraction, such as moving average or else higher order polynomials that more accurately model the baseline changes in their data.So that readers can judge the fidelity of baseline subtraction for themselves, perhaps the authors should present all (not just representative) raw bioluminescence traces as supplementary data along with the baseline they have subtracted in each case.

This series of questions and prompts became a major topic of investigation and study. These comments were helpful as they directed us to specific topics rather than simply stating that the data analysis needed improvement. With serious study, experimentation, and helpful discussion from Dr. David Ferster of Actimetrics we began to understand the critical importance of baseline subtraction. As an example, we present analysis of the current Figure 2. Using wild type and *Nrf2-/-* MEFs, these data demonstrate that Nrf2 is required for normal circadian timekeeping.

We will illustrate the effect of baseline correction with Figure 2. The figure now included in the manuscript is corrected with a 24 hour running average baseline subtraction.

The data now presented in Figure 2—figure supplement 1 shows all of the raw data for Figure 2. In the lower panel, we have enhanced the scale of the y-axis to demonstrate the quality of the rhythm in the Nrf2-null mouse embryonic fibroblasts (MEFs). The insert (right graph) is further enhanced for days 2-6.

Author response image 1 shows four different baseline corrections of this data, the top three, using polynomials 1, 2, and 3, and a 24 hour running average baseline subtraction.

Author response image 2 shows the goodness-of-fit from the preceding analyses. It is obvious that the 24 hour running average baseline subtraction provides the best goodness of fit and can be used with the different cell lines without altering the parameters for each individual cell line. Further, we found that our previous analysis of period length was very similar to this new output and that amplitude was also similar with a slightly higher variance. Thus, we re-analyzed all of our data using a 24 hour running average baseline subtraction method and were able to substantiate our earlier findings with an improved fit of the data.

**Author response image 2. respfig2:** 

Raw Bioluminescence data is now shown as child figures to all key experiments. These include: Figure 2, Figure 2—figure supplement 1; Figure 2, Figure 2—figure supplement 2; Figure 4, Figure 4—figure supplement 1, Figure 4—figure supplement 3; Figure 4, Figure 4—figure supplement 4; Figure 4, Figure 4—figure supplement 5.

Changes in amplitude/luciferin concentration:The apparent amplitude, phase and period of circadian rhythms reported bioluminescently is quite sensitive to the kinetics of luciferin transport over the plasma membrane, unless intracellular luciferin concentration is in saturating excess (Patrick et al., 2014; Feeney et al., 2016). Please could the authors report explicitly the concentration of extracellular luciferin employed during their assays and confirm that this is in excess i.e. addition of more luciferin extracellularly will elicit no further increase in bioluminescence if luciferin is not rate-limiting for enzyme activity.

The concentration of luciferin is 1mM.

Biological effect size:Possibly related to the above points, I note that the difference in circadian period between wild type cells between experiments was as much as 2 hours (compare Figure 2 Wt with Figure 2 Wt or any Wt controls in Figure 4, for example). Given this large σ, a power calculation indicates that n=3 should be underpowered to detect small differences in circadian period as reported in Figure 4 for example. Could the authors please report in the methods section whether experiments that showed small (<1 h), but significant, differences in period were repeated on a separate occasion with the same result.

The reviewers can now review the raw and processed data for multiple key experiments described above. Because of difference in bioluminescence scale over long periods of time such as year to year, we chose to perform replicated experiments over a period of weeks to months. As can be seen in the data presented, the experimental replication is quite strong. Further, we repeated each experiment as a biological replicate of the effect. In addition, we performed experiments that were reinforcing so that we could build a strong case for consistency of biological effect. For example, the difference between wild-type and Nrf2-null MEFs is replicated in six different experiments included in the manuscript. In addition, this result is further supported in 4 shRNA experiments in two different cell lines. Similarly, the gains of function phenotypes are demonstrated using multiple chemical activators of Nrf2, overexpression of Nrf2 by recombinant means, and genetically in Keap1-null cells. Finally, the animal experiments were replicated in two independent strains of mice produced by separate labs. Further, nearly every experiment measures both levels of Nrf2 protein and also the level of Nqo1, a gene transcriptionally regulated by Nrf2. Thus, we believe that all presented results are strongly supported by independent analyses of many kinds.

I also noticed that large differences in amplitude are also observed between experiments for Wt controls (see Figure 2 for example). Given that the wild type controls in 2A vs. 2B are effectively experimental repeats of each other, I do not understand why the amplitude differs by an order of magnitude. Could the authors please comment?

The experiments reported in Figure 2 were performed over a year apart. As our cells were not cloned, drift is a possibility. Other more simple types of issues could arise from how well the cells were frozen or grown, or simply instrumental variability between the two LumiCycle instruments in the lab.

[Editors' note: further revisions were requested prior to acceptance, as described below.]

Thank you for resubmitting your work entitled "NRF2 regulates core and stabilizing circadian clock loops, coupling redox and timekeeping in Mus musculus" for further consideration at eLife. Your revised article has been favorably evaluated by Philip Cole (Senior editor), a Reviewing editor, and two reviewers.This revised manuscript is an enormous improvement on the original submission and should be accepted for publication pending addressing two remaining issues. There are two places where the description of experimental findings might be seen as slightly misleading:1) Taken in isolation, the Abstract suggests that the authors have identified a general NRF2-dependent mechanism by which redox balance is communicated to modulate clock gene expression rhythms. This is not quite true, since there was no effect of NRF2 deletion on lung or SCN explants. It would be less misleading to add "in some cell types" to the end of the third sentence of the Abstract. This in no way diminishes these important findings and is a more accurate summation of their findings.2) The penultimate sentence of the discussion is heavily metaphor laden and could readily be misconstrued by readers outside the circadian field. Both of the terms "transcriptional clock" and "metabolic redox oscillator" suggest self-sufficiency – which is unsubstantiated. Again, it would be more accurate and lucid to amend to something like: "…NRF2 appears to be a key mechanistic link between circadian oscillations in redox balance and clock gene expression rhythms."

Thank you for the favorable review. We have made the changes suggested by the reviewers. We believe that in doing so we have enhanced the clarity of the manuscript. We never strive to mislead a reader.